# CODiT: Conformal Out-of-Distribution Detection in Time-Series Data

## Abstract

Machine learning models are prone to making incorrect predictions on inputs that are far from the training distribution. This hinders their deployment in safety-critical applications such as autonomous vehicles and healthcare. The detection of a shift from the training distribution of individual datapoints has gained attention. A number of techniques have been proposed for such out-of-distribution (OOD) detection. But in many applications, the inputs to a machine learning model form a temporal sequence. Existing techniques for OOD detection in time-series data either do not exploit temporal relationships in the sequence or do not provide any guarantees on detection. We propose using deviation from the in-distribution temporal equivariance as the non-conformity measure in conformal anomaly detection framework for OOD detection in time-series data. Computing independent predictions from multiple conformal detectors based on the proposed measure and combining these predictions by Fisher's method leads to the proposed detector CODiT with guarantees on false detection in time-series data. We illustrate the efficacy of CODiT by achieving state-of-the-art results on computer vision datasets in autonomous driving. We also show that CODiT can be used for OOD detection in non-vision datasets by performing experiments on the physiological GAIT sensory dataset. Code, data, and trained models are available at `https://drive.google.com/file/d/1uSNYGoSNWu4_d-nPAcIcxYyBYI8VLqPq/view?usp=sharing`.

## 1 Introduction

Deployment of machine learning models in safety-critical systems such as autonomous driving (Bojarski et al., 2016), and healthcare (De Fauw et al., 2018) is hindered by uncertainty in the outputs of these models. One such source of uncertainty at the inference time is a shift in the distribution of the model's inputs from their training distribution. Detection of this shift or out-of-distribution (OOD) detection on individual datapoints has therefore gained attention (Hendrycks & Gimpel, 2016; Lee et al., 2017; 2018; Hendrycks et al., 2019; Tack et al., 2020; Kaur et al., 2021). But this problem of OOD detection in time-series data is less explored (Cai & Koutsoukos, 2020; Ramakrishna et al., 2021; Feng et al., 2021). In time-series data, OOD detection aims at detecting those windows of time-series datapoints that are outside the training distribution. In this case, the distribution of interest is not just of individual datapoints, but the distribution of sequences in which these datapoints occur in the time-series data. In this paper, we propose CODiT, a novel algorithm for OOD detection in time-series data with a bounded false detection rate.

Most of the existing approaches for detection in time-series data are *point-based*, i.e., they independently consider each datapoint in the window, such as individual frames in a video clip. In other words, these approaches do not exploit time-dependency among the datapoints in a window to detect if the window is OOD. An example of an OOD window is a car drifting video clip due to slippery ground or loss of control, given normal driving video clips as in-distribution (iD) data. As shown in Fig. 1, we cannot tell from a single frame if the car has drifted since drift is defined based on the relative relation between frames (e.g., a trajectory of a car). We define these OOD windows as *temporal OODs*, which require considering the sequence of datapoints in the window for detection. In contrast to the existing point-based approaches, we propose using time-dependency among the datapoints in a window for detection of temporal OODs. Specifically, we propose *using deviation from the iD temporal equivariance*, i.e. equivariance with respect

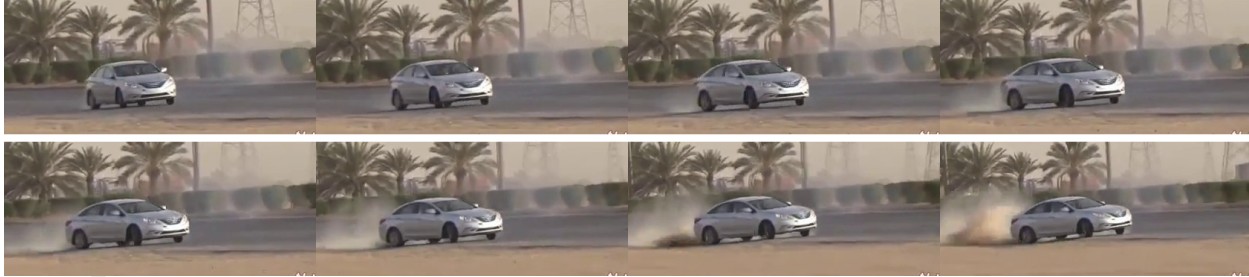

Figure 1: Drift in cars as temporal OODs. This trace is taken from the drift dataset by Noor et al. (2020).

Table 1: Detection capabilities of OOD detectors in time-series data.

| Detector | False Detection Rate Guarantees | Temporal OODs | Non-vision Datasets |
|---|---|---|---|
| VAE-based (Cai & Koutsoukos, 2020) | ✓ | ✗ | ✓ |
| $\beta$ VAE-based (Ramakrishna et al., 2021) | ✓ | ✗ | ✓ |
| Optical Flow based (Feng et al., 2021) | ✗ | ✓ | ✗ |
| CODiT (Ours) | ✓ | ✓ | ✓ |

to a set of temporal transformations learned by a model on windows drawn from the training distribution, *for OOD detection in time-series data*. This is because a model trained to learn equivariance with respect to temporal transformations on data drawn from the training distribution might not generalize or exhibit equivariance on OOD samples dissimilar to the training distribution.

To bound false detection on the iD data, we leverage inductive conformal anomaly detection (ICAD) (Balasubramanian et al., 2014). ICAD is a general framework for testing if an input conforms to the training distribution by computing quantitative scores defined by a non-conformity measure. A $p$-value, computed by comparing non-conformity score of the input with these scores on the data drawn from training distribution, indicates anomalous behavior of the input. The detection performance of ICAD, however, depends on the choice of the non-conformity measure used in the framework (Balasubramanian et al., 2014). We propose using *deviation from the expected temporal equivariant behavior of a model learned on data drawn from the training distribution* as the non-conformity measure in ICAD for OOD detection in the time-series data.

ICAD computes a single $p$-value of the input to detect its anomalous behavior. To enhance detection performance, we propose using multiple ($n > 1$) $p$-values computed from $n$ transformations sampled as independent and identically distributed (IID) variables from a distribution over the set of temporal transformations. The intuition for using multiple transformations is that an OOD window might behave as a transformed iD window with one transformation but the likelihood of this decreases with the number of temporal transformations. Using Fisher's method (Toccaceli & Gammerman, 2017) to combine these $n$ independent $p$-values leads to the proposed detector CODiT for conformal OOD detection in time-series data with a bounded false detection rate. The contributions of this paper can be summarized as:

**1. Novel Measure for OOD Detection in Time-Series Data.** To our knowledge, all the existing nonconformity measures for OOD detection are defined on individual datapoints. We propose a measure that is defined on the window containing information about the sequence of time-series datapoints for enhancing the detection of temporal OODs. With a model trained to learn iD temporal equivariance via the auxiliary task of predicting an applied transformation on windows drawn from the training distribution, we propose using error in this prediction as the non-conformity measure in ICAD for OOD detection in time-series data.
**2. Enhanced detection performance.** To enhance the detection performance, we propose to use Fisher's method as an ensemble approach for combining predictions from multiple conformal anomaly detectors based on the proposed measure.
**3. CODiT.** Computing $n$ independent $p$-values of the input from the proposed measure in the ICAD framework, and combining these values by Fisher's method leads to the proposed detector CODiT with a bounded false detection rate.
**4. Evaluation.** For comparison with the point-based approaches, we perform experiments on weather and night OODs on a driving dataset simulated by CARLA (Dosovitskiy et al., 2017), achieving state-of-the-art (SOTA) results. We outperform the existing non-point based SOTA (Feng et al., 2021) on vision temporal OODs. To illustrate that CODiT can be used for OOD detection beyond vision, we also perform experiments and obtain SOTA results on the real physiological GAIT dataset (Hausdorff et al., 2000).

## 2 Problem statement and Motivation

### 2.1 Problem Statement

OOD detection in time-series data takes in a window $X_{t,w}$ of consecutive time-series datapoints $(x_t, x_{t+1}, \ldots, x_{t+w-1})$, and labels $X_{t,w}$ as iD or OOD. Here $t$ is the starting time of the window and $w$ is the window length.

### 2.2 Motivation of the Proposed OOD Detection Measure

The existing point-based detectors might not be able to detect temporal OODs. An example of a temporal OOD, as shown in Fig. 2, is the replay window where camera gets stuck at a single frame and starts generating the same image over and over again. We need to consider the sequence of same image in a replay window to detect the window as OOD. Detection results on replay dataset in Fig. 3 shows that both the existing point-based detectors, namely variational autoencoder (VAE) based Cai et al.'s (2020), and $\beta$ VAE-based Ramakrishna et al.'s (2021), perform poorly in the detection of these temporal OODs. We, therefore, propose using time-dependency among the datapoints in a window for OOD detection in time-series data.

To our knowledge, Feng et al.'s (2021) detector is the only existing OOD detector in time-series data that takes into account time-dependency among the individual datapoints in a window. It does so by extracting optical flow information from consecutive frames in a video clip. As shown in the experimental results on temporal OODs (Fig. 7) of Section 5, Feng et al.'s detector can thus be used to detect temporal OODs. However, since this detector depends on the optical flow information, it is restricted to the vision data. In contrast, CODiT can be used to detect temporal OODs across domains without relying on any domain-specific features.

Table 1 compares the detection capabilities of CODiT with the existing OOD detectors in time-series data.

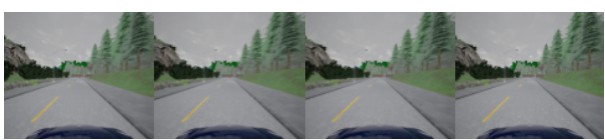

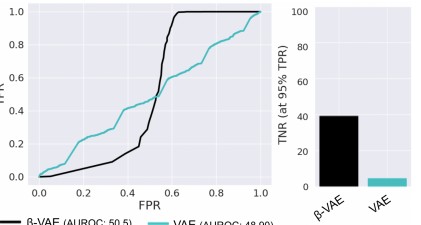

Figure 2: **Replay window**: An example of the temporal OOD where camera gets stuck and generates the same image in the entire window. This trace is generated by CARLA, an open-source simulator for autonomous driving research (Dosovitskiy et al., 2017).

Figure 3: **ROC curves (left), TNR (at 95% TPR on right) results on replay OODs.** The existing point-based OOD detectors in time-series data, namely VAE-based Cai et al.'s (2020), and $\beta$ VAE-based Ramakrishna et al.'s (2021) perform poorly in the detection of these temporal OODs.

## 3 Background and Notations

CODiT uses error in the temporal equivariance learned by a model on windows drawn from the training distribution as the non-conformity measure in the inductive conformal anomaly detection (ICAD) framework. With multiple $p$-values obtained from the proposed measure in ICAD, the final OOD detection score is computed by combining these values by Fisher's method. Here we provide the background on equivariance, ICAD, and Fisher's method required for technical details of the proposed OOD detector, CODiT. We also define the notations used in the rest of the paper.

### 3.1 Equivariance

A function $f$ is equivariant with respect to a transformation $g$ if we know how the output of $f$ changes if we transform its input from $x$ to $g(x)$. Learning features that are equivariant to translations on inputs by sharing of kernels in the convolutional neural networks has played a crucial role in the success of these networks (Cohen & Welling, 2016). Invariance is a special case of equivariance where the output of $f$ does not change by the transformation $g$ on its input. Invariance with respect to geometric transformations such

as rotation, tilt, scale, etc. is a desired property of the deep learning classifiers. For example, classification results on the upright images of cats should not change with a tilt in these images.

**Definition 1** (Schmidt & Roth, 2012). *For a set $X$, a function $f$ is defined to be equivariant with respect to a set of transformations $G$, if there exists the following relationship between any transformation $g \in G$ of the function's input and the corresponding transformation $g'$ of the function's output:*

$$f(g(x)) = g'(f(x)), \forall x \in X. \tag{1}$$

Invariance is a special case of equivariance where $g'$ is the identity function, i.e., the output of $f$ remains unchanged by the transformation $g$ on its input.

**Learning Equivariance: Autoencoding Variational Transformations (AVT).** Augmenting training data with transformations from the set $G$ of geometric transformations is a common approach to learning invariance with respect to $G$ (Chen et al., 2020a;b; Chatzipantazis et al., 2021). The auxiliary task of predicting an applied transformation from $G$ on the training data also encourages the model to learn equivariance with respect to $G$ (Qi et al., 2019). Qi et al.'s 2019 "Autoencoding Variational Transformations" (AVT) framework trains a VAE to learn a latent space that is equivariant to transformations. For the set $X$ of training images and the set $G$ of geometric transformations, a VAE is trained to predict the applied transformation from $G$ on an input $x \in X$. Equivariance between the latent space of VAE and $G$ is learned by maximizing mutual information between the latent space and $G$.

**Definition 2** (Jenni & Jin, 2021). *Temporal equivariance of a function $f$ from equation 1 is defined on a set $X$ of windows of consecutive time-series datapoints and with respect to a set $G$ of temporal transformations.*

Some examples of temporal transformations on video clips are skipping every second frame in the clip (2x speed), shuffling the frames in the clip (shuffle), reversing the order of frames in the clip (reverse), and reversing the order of the second half frames in the clip (periodic).

In the rest of the paper, we will use the notation $G_T$ to represent a set of temporal transformations. We call a function $f$ as $G_T$-equivariant if it learns equivariant representations of windows drawn from the training distribution with respect to the set $G_T$ of temporal transformations. We refer to the deviation from the expected result of this function $f$ on a transformed input (transformed with a $g \in G_T$) as deviation from the iD $G_T$-equivariance.

### 3.2 Inductive Conformal Anomaly detection (ICAD)

Inductive Conformal Anomaly Detection (ICAD) (Laxhammar & Falkman, 2015) is a general framework for testing if an input conforms to the training distribution. It is based on a non-conformity measure (NCM), which is a real-valued function that assigns a non-conformity score $\alpha$ to the input. This score indicates non-conformance of the input with data drawn from the training distribution. The higher the score is, the more non-conforming or anomalous the input is with respect to training data. An example of the non-conformity score is the reconstruction error by a VAE trained on data drawn from the training distribution.

The training dataset $X$ of size $l$ is split into a *proper training set* $X_{\mathrm{tr}} = \{x_j : j = 1, \ldots, m\}$ and a *calibration set* $X_{\mathrm{cal}} = \{x_j : j = m+1, \ldots, l\}$. Proper training set $X_{\mathrm{tr}}$ is used in defining NCM. In the example of reconstruction error by a VAE as the non-conformity score, the VAE trained on $X_{tr}$ is used for computing the error. Calibration set $X_{\mathrm{cal}}$ is a held-out training set that is used for computing $p$-value of an input. $p$-value of an input $x$ is computed by comparing its non-conformity score $\alpha(x)$ with these scores on the calibration datapoints:

$$p\text{-}value(x) = \frac{|\{j = m+1, ..., l : \ \alpha(x) \le \alpha(x_j)\}| + 1}{l - m + 1}. \tag{2}$$

If $x$ is drawn from the training distribution, then its non-conformity score is expected to lie within the range of scores for the calibration datapoints and thus higher $p$-values for the iD datapoints. With $\epsilon \in (0, 1)$ as the anomaly detection threshold, $x$ is therefore detected as an anomalous input if the $p$-value of $x$ is less than $\epsilon$.

**False Detection Rate Guarantees.** The false anomalous detection on an input drawn from the training distribution is upper bounded by the specified detection threshold $\epsilon$ in the ICAD framework.

**Lemma 1** (Balasubramanian et al., 2014). *If an input $x$ and the calibration datapoints $x_{m+1}, \ldots, x_l$ are independent and identically distributed (IID), then for any choice of the NCM defined on the proper training set $X_{tr}$, the p-value(x) in equation 2 is uniformly distributed. Moreover, we have $Pr\left(p\text{-}value(x) < \epsilon\right) \leq \epsilon$, where the probability is taken over $x_{m+1}, \ldots, x_l$, and $x$.*

From Lemma 1, we know that if $x$ and the datapoints in the calibration set $X_{\text{cal}}$ are IID, then the $p$-value(x) from equation 2 is uniformly distributed over $\{1/(l-m+1), 2/(l-m+1), \ldots, 1\}$. The probability of $p$-value(x) less than $\epsilon$ or misclassifying $x$ as anomalous is, therefore, $\sum_{1 \leq i \leq (l-m+1)\epsilon} 1/(l-m+1) = \lfloor(l-m+1)\epsilon\rfloor/(l-m+1) \leq \epsilon$.

### 3.3 Fisher's Method

The same hypothesis can be tested by multiple conformal predictors and an ensemble approach for combining these predictions can be used to improve upon the performance of individual predictors. Fisher's method is one of these approaches for combining multiple conformal predictions or $p$-values of an input from equation 2. Fisher value of an input $x$ from $n$ $p$-values is computed as follows:

$$\text{fisher-value}(x) = t \sum_{i=0}^{n-1} \frac{(-\log t)^i}{i!}, \text{ where } t = \prod_{k=1}^{n} p_k. \tag{3}$$

**Lemma 2** (Toccaceli & Gammerman, 2017). *If $n$ p-values, $p_1, \ldots, p_n$, are independently drawn from a uniform distribution, then $-2\sum_{i=1}^{n} \log p_i$ follows a chi-square distribution with $2n$ degrees of freedom. Thus, the combined p-value is*

$$Pr\left(y \leq -2\sum_{i=1}^{n} \log p_i\right) = t\sum_{i=0}^{n-1} \frac{(-\log t)^i}{i!},$$

*where $t = \prod_{k=1}^{n} p_k$, $y$ is a random variable following a chi-square distribution with $2n$ degrees of freedom, and the probability is taken over $y$. Moreover, the combined p-value follows the uniform distribution.*

## 4 Temporal Equivariance for Conformal OOD Detection in Time-Series Data

Here, we first classify OOD data in time-series into temporal and non-temporal types, and then provide details of the proposed detector CODiT.

### 4.1 OOD Data Types in Time-Series

We classify OOD windows in time-series data into two types: *temporal* OODs and *non-temporal* OODs.

A crucial property of the temporal OODs compared to the non-temporal OODs is that it is hard to detect temporal OODs by looking at individual datapoints within the window without considering time-dependency between these datapoints. Examples of temporal OODs in autonomous driving are car drifting video clips (Fig. 1), and replay OODs (Fig. 2). An example of the temporal OOD in healthcare is the GAIT (or waking pattern) of patients with neurodegenerative diseases. With the GAIT of healthy individuals as iD data, the GAIT of patients with neurodegenerative diseases, such as Parkinson's disease (PD), Huntington's disease (HD), and Amyotrophic Lateral Sclerosis (ALS), are examples of temporal OODs. Fig. 4 shows dynamics of the stride time (one of the walking

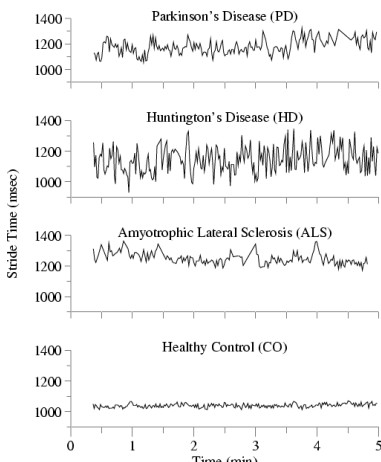

Figure 4: GAIT in patients with neurodegenerative diseases (Hausdorff et al., 2000) as temporal OODs.

pattern features) of a healthy control person and patients with PD, HD, and ALS disease. As shown in the Figure, we need a sequence of time-series datapoints to determine whether the walking pattern is from a healthy individual or a patient. In contrast to the temporal OODs, the non-temporal OODs can be detected by looking at individual datapoints. Examples of non-temporal OODs include driving video clips under rainy, foggy, or snowy weather, given the driving video clips under clear sunny weather as iD data. We can detect weather OODs by looking at images in the window independently.

Based on these observations, we call a window $X_{t,w}$ as a temporal OOD if $X_{t,w}$ is drawn from OOD but confused to be drawn from iD by removing the time-dependency of individual datapoints within $X_{t,w}$ (e.g., randomly shuffling the order of video clip frames). As shown in the experimental section 5, CODiT can be used to detect both temporal and non-temporal OODs in time-series data.

## 4.2 CODiT

CODiT uses an OOD detection score based on multiple $p$-values from ICAD. Here, we first define the proposed NCM to be used in the ICAD framework for computing a $p$-value along with the final detection score, and then formalize CODiT's algorithm with a bounded false detection rate.

### 4.2.1 Proposed NCM and OOD Detection Score

**Proposed NCM.** We propose to use time-dependency between datapoints in a time-series window for detection on the window. Unlike all the existing NCMs defined on individual datapoints, we propose an NCM that is defined on the window containing information about the sequence of datapoints in the window. Specifically, we propose using deviation from the expected iD $G_T$-equivariance learned by a model on windows drawn from the training distribution as an NCM in ICAD for OOD detection in time-series data. Learning $G_T$-equivariance via an auxiliary task of predicting the applied temporal transformation (such as shuffle, reverse, etc.) on a window requires learning changes in the original sequence of the datapoints in a predictable way. For a VAE model $M$ trained to learn $G_T$-equivariance on windows of proper training data in the AVT framework, we propose to use error in the prediction of the applied temporal transformation $g \in G_T$ on an input window $X_{t,w}$ as the NCM: **PredictionError$(\mathbf{g}, \mathbf{M}(\mathbf{g}(\mathbf{X_{t,w}})))$**.

We call the proposed NCM as the **Temporal Transformation Prediction Error (TTPE)** NCM.

The existing AVT framework (Qi et al., 2019) is defined to learn equivariance with respect to geometric transformations on images. We extend it to learn $G_T$-equivariance by:

(1) Modifying VAE's architecture to accept windows of consecutive time-series datapoints as inputs. The time-series can be on vision (e.g. drift car video clip) or non-vision (e.g., GAIT) datapoints.
(2) Modifying the auxiliary task to predict the applied temporal transformation from a set $G_T$ on windows of time-series datapoints.

**Motivation for TTPE-NCM.** $G_T$-equivariance learned by a model on windows drawn from the training distribution is more likely to work on iD data and is not guaranteed to generalize to OOD data dissimilar to that used for training. With the set $G_T = \{2\text{x } speed, shuffle, reverse, periodic, identity\}$, we train a VAE model on the proper training data of the drift dataset to predict an applied transformation $g$ sampled independently from a uniform distribution over $G_T$. With $G_T$ as the set of five classes of temporal transformations, we use CrossEntropyLoss$(g, M(g(X_{t,w})))$ as the PredictionError$(g, M(g(X_{t,w})))$. Fig. 5 shows that the model has much higher prediction losses on the OOD windows than on the test iD windows on all the five ground truth transformations in $G_T$. This supports our hypothesis that $G_T$-equivariance learned on data drawn the training distribution is not likely to generalize on data drawn from OOD, and therefore higher prediction errors on OOD windows than on the iD windows.

**OOD Detection Score.** Instead of using a single $p$-value from the TTPE-NCM in ICAD, we propose using multiple ($n > 1$) $p$-values to enhance detection. We require $n$ non-conformity scores for both the input and the calibration datapoints for computing $n$ $p$-values. These scores are computed from $n$ transformations sampled independently from a distribution $Q_{G_T}$ over $G_T$ for both the input and the calibration datapoints:

$$\alpha_i(X_{t,w}) = \text{PredictionError}(g_i, M(g_i(X_{t,w}))) \ : \ 1 \leq i \leq n, g_i \sim Q_{G_T},$$

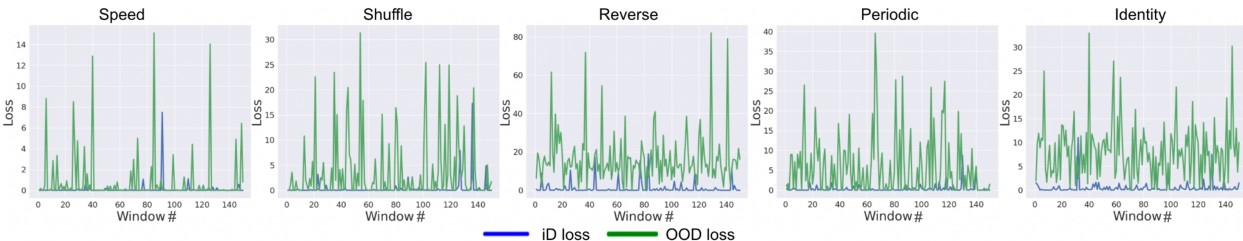

Figure 5: Higher $Prediction\ Error(g, M(g(X_{t,w}))) = \text{CrossEntropyLoss}(g, M(g(X_{t,w})))$ on the OOD windows than on the test iD windows of the drift dataset. This shows that $G_T$-equivariance learned on the windows drawn from the training distribution is less likely to generalize on the windows drawn from OOD.

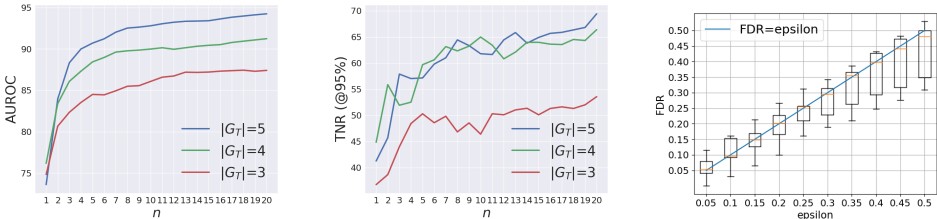

Figure 6: AUROC vs. $n$ (left), TNR vs. $n$ (center) shows that the performance of CODiT increases with the increase in the number $n$ of $p$-values used in the fisher-value for detection. False Detection Rate (FDR) of CODiT ($n = 5$) is bounded by $\epsilon$ on average (right).

where $X_{t,w}$ is the input or a calibration datapoint. Using Fisher's method to combine these $n$ $p$-values gives us the fisher-value of input from equation 3. This value is expected to be higher for iD datapoints than OOD datapoints (Fisher, 1932), and therefore we perform detection by using a threshold on the fisher-value of input. In other words, CODiT uses fisher-value of the input as the final OOD detection score.

**Motivation for multiple $p$-values for OOD Detection.** A single $p$-value measures deviation from the iD $G_T$-equivariance of the input with respect to one transformation $g \sim Q_{G_T}$. With multiple $p$-values, we test this deviation with respect to multiple transformations sampled independently from $Q_{G_T}$. We hypothesize that under one transformation, an OOD window might behave as the transformed iD window but the likelihood of this decreases with the number of transformations. For testing this hypothesis, we train three VAE models with the set $G_T$ equal to $\{2\text{x } speed, reverse, identity\}$, $\{2\text{x } speed, shuffle, periodic, identity\}$, and $\{2\text{x } speed, reverse, shuffle, periodic, identity\}$, respectively. These models are trained on the proper training set of the drift dataset to predict an applied transformation $g$ sampled independently from a uniform distribution over $G_T$. Again, we use $\text{PredictionError}(g, M(g(X_{t,w})))$ as the $\text{CrossEntropyLoss}(g, M(g(X_{t,w})))$. Fig. 6 shows that the detection performance (in AUROC and TNR) of CODiT increases as we increase the number $n$ of $p$-values used in the final OOD detection score (or the fisher-value) for all of the three cases ($|G_T| = 3,\ 4,$ and $5$). This supports our hypothesis on using multiple $p$-values for enhancing detection.

### 4.2.2 Algorithm and Guarantee for OOD Detection

For the ICAD guarantees from Lemma 1 to hold on a $p$-value for an input sampled from the training distribution, we require the calibration set used in the $p$-value computation to be IID. With time-series calibration traces, we propose to create an IID calibration set by using exactly one calibration window from each calibration trace. For each trace, this window is sampled independently from a uniform distribution over the calibration windows in the trace. With $Q_{G_T}$ as the distribution over the set $G_T$ of temporal transformations, non-conformity scores on the calibration windows in the set are computed from the TTPE-NCM by sampling a transformation independently from $Q_{G_T}$ for each window in the set. $n$ such sets of non-conformity scores on the $n$ IID calibration sets are computed and passed as an input to the proposed Algorithm 1 for OOD detection in time-series data.

Line 4 of the Algorithm samples a transformation $g$ independently from $Q_{G_T}$. The transformed input $g(X_{t,w})$ is passed through the VAE model $M$ trained to learn $G_T$-equivariance on the windows drawn from the proper training data (Line 5). Line 6 computes the non-conformity score $\alpha$ of $X_{t,w}$ from the TTPE-NCM, which is the prediction error function $f$ over the applied transformation $g$ on $X_{t,w}$ and the transformation $\hat{g}$ predicted

by $M$. $p$-value of $X_{t,w}$ is computed in Line 7 by comparing its non-conformity score with these scores on the calibration windows. This process from sampling a transformation from $Q_{G_T}$ to computing the $p$-value of $X_{t,w}$ is repeated $n$ times to compute $n$ $p$-values of $X_{t,w}$ (Lines 3 to 8). $X_{t,w}$ is detected as OOD if the fisher-value computed from its $n$ $p$-values is less than the desired false detection rate $\epsilon$ (Line 10).

---

**Algorithm 1** CODiT: Conformal Out-of-Distribution Detection in Time-Series Data

---

1: **Input:** a window $X_{t,w}$ of time-series data, VAE model $M$ trained on proper training set of the iD windows, distribution $Q_{G_T}$ over the set $G_T$ of temporal transformations, prediction error function $f$, $n$ sets of calibration set alphas $\{\alpha_j^k : 1 \le k \le n, m+1 \le j \le l\}$, and a desired false detection rate $\epsilon \in (0, 1)$

2: **Output:** "1" if $X_{t,w}$ is detected as OOD; "0" otherwise
3: **for** $k \leftarrow 1, \ldots, n$ **do**
4:     $g \sim Q_{G_T}$
5:     $\hat{g} \leftarrow M(g(X_{t,w}))$
6:     $\alpha \leftarrow f(g, \hat{g})$
7:     $p_k \leftarrow \frac{|\{j=m+1,\ldots,l:\ \alpha \le \alpha_j^k\}|+1}{l-m+1}$
8: **end for**
9: $t \leftarrow \prod_{k=1}^{n} p_k$
10: **if** $t \sum_{i=0}^{n-1} \frac{(-\log t)^i}{i!} < \epsilon$ **then return** 1 **else return** 0

---

**Theorem 1.** *The probability of false OOD detection on $X_{t,w}$ by Algorithm 1 is upper bounded by $\epsilon$.*

*Proof.* An IID calibration set is used for a $p$-value computation in Algorithm 1. If an input $X_{t,w}$ is sampled from the training distribution, then $X_{t,w}$ and datapoints in the calibration set are also IID. The non-conformity scores of $X_{t,w}$ and calibration datapoints used in the $p$-value computation of Line 7 of the Algorithm 1 are therefore IID conditioned on the proper training set and the set of temporal transformations $G_T$. With the $n$ IID calibration sets sampled independently from the calibration traces, $n$ non-conformity scores computed from $n$ transformations sampled independently from $Q_{G_T}$ for both the input and the calibration datapoints, and Lemma 1, the $n$ $p$-values of $X_{t,w}$ computed in Algorithm 1 are independent and uniformly distributed. Due to this property on the $n$ $p$-values and Lemma 2, the combined $p$-value in Line 10 of Algorithm 1 is also uniformly distributed. Therefore, the probability of falsely detecting $X_{t,w}$ as OOD from the combined $p$-value (or the fisher-value($X_{t,w}$)) is upper bounded by $\epsilon$ due to Lemma 1. □

The unconditional probability that an input $X_{t,w}$ sampled from the training distribution $D$ is classified as OOD by Algorithm 1 is bounded by $\epsilon$. For this guarantee to hold for a sequence of inputs, we require an independent calibration set for every input in the sequence. This is computationally inefficient for real-time applications and therefore a fixed calibration set is used for all the inputs in the offline version of the ICAD algorithm (Laxhammar & Falkman, 2015). The average false detection rate on the sequence of inputs drawn from $D$ in this setting is expected to be empirically calibrated with or even higher than $\epsilon$. We also fix the $n$ sets of IID calibration datapoints and pass it as an input to the Algorithm 1. Box plots in Fig. 6 (right) show that the false detection rate of CODiT is empirically bounded by $\epsilon$ on average.

## 5 Experimental Results

We perform experiments on the following three computer vision datasets in autonomous driving: (Dataset 1) a driving dataset under different weather conditions generated by CARLA, an open-source simulator for autonomous driving research (Dosovitskiy et al., 2017); (Dataset 2) a replay OOD dataset simulated by CARLA; and (Dataset 3) a real driving drift dataset (Noor et al., 2020). In addition, to illustrate that CODiT can be used for OOD detection beyond vision, we also perform experiments on: a real physiological GAIT sensory dataset (Hausdorff et al., 2000) (Dataset 4).

All the existing approaches report their results on weather OODs. So, we compare CODiT's performance with the existing approaches on weather OODs (Dataset 1). Results on temporal OODs in vision datasets, i.e., replay (Dataset 2), and drift (Dataset 3) are compared with the existing non-point based SOTA (Feng et al., 2021). Since Feng et al. (2021)'s approach is not applicable to non-vision datasets, we generate a

Table 2: Comparison of CODiT with Cai et al.'s (VAE), Ramakrishna et al.'s ($\beta-$VAE), and Feng et al.'s detectors on weather and night OODs from CARLA dataset. Best results are in bold.

| OOD | AUROC (↑) | | | | TNR (90% TPR) (↑) | | | | Detection Delay (@95% TPR) (↓) | | | |
|---|---|---|---|---|---|---|---|---|---|---|---|---|
| | VAE | $\beta-$VAE | Feng's | Ours | VAE | $\beta-$VAE | Feng's | Ours | VAE | $\beta-$VAE | Feng's | Ours |
| Rainy | 53.56 | 92.07 | 84.21 | **99.71** | 0 | 81.00 | 27.63 | **98.57** | NA | **0.15** | 5.33 | **0.15** |
| Foggy | 52.02 | 41.02 | 86.09 | **99.66** | 2.30 | 2.75 | 28.01 | **98.05** | 33.05 | 19.65 | 5.37 | **0** |
| Snowy | 53.23 | **97.52** | 95.91 | 96.67 | 0 | **99.69** | 78.20 | 86.47 | NA | 0.33 | **0** | **0** |
| Night | 50.86 | 95.57 | 75.07 | **98.94** | 1.78 | 71.90 | 0.40 | **94.55** | 72.80 | 4.07 | 85.4 | **1.41** |

non-point based baseline for the detection of temporal OODs in GAIT Dataset 4 and compare our results with it. We also perform ablation studies on the drift dataset.

## 5.1 Weather OODs

### 5.1.1 Dataset

**Training Set.** We generate 33 driving traces of varying lengths in clear daytime weather as the iD training traces. We randomly split these into 20 traces of the proper training set $X_{tr}$ and 13 traces of the calibration set $X_{cal}$. Windows from $X_{tr}$ are sampled for training the model. Windows from $X_{cal}$ are sampled $n = 20$ times (with one window from each calibration trace at a time to make each window in the set independent) for calculating the 20 sets of calibration non-conformity scores.

**Test Set.** We generate 27 driving traces of varying lengths on a clear day weather as the iD test traces. Weather and night time OOD traces are generated by using the automold software (Saxena, 2018)[1] on the 27 iD test traces. OOD traces start from iD and gradually become OOD, i.e., the intensity of rain, fog, snow, or low brightness (for night) starts increasing gradually turning into the OOD traces. Examples of these iD and OOD windows are shown in Appendix.

### 5.1.2 Training Details, $G_T$, and TTPE-NCM

We train an VAE model with the R3D network architecture (Tran et al., 2018) on the windows of length $w = 16$ from $X_{tr}$. R3D network is the 3D CNNs with residual connections and thus can be used on the 3D time-series input data. We use $G_T = \{$2x Speed, Shuffle, Periodic, Reverse, Identity$\}$ and train the model to predict the applied transformation $g \in G_T$ with cross-entropy loss between the true and the predicted transformation. CrossEntropyLoss$(g, M(g(X_{t,w})))$ is used as the *Prediction Error*$(g, M(g(X_{t,w})))$. The value of this loss is used as the non-conformity score $\alpha$ for computing the $p$-value of an input $X_{t,w}$.

### 5.1.3 Results

We report results on the sliding windows ($w = 16$) of the test iD and OOD traces. We call iD as positive and OOD as negative. We report AUROC, TNR(@95% TPR), and detection delay(@95% TPR) in Table 2. Starting from the first ground truth OOD window in a trace, number of windows required to detect the first OOD window in the trace is reported as the detection delay. This number is averaged over the total number of OOD traces and reported in the table. As can be seen CODiT outperforms other approaches, except for Snowy OOD, where our approach is the second best.

## 5.2 Temporal OODs

### 5.2.1 Vision: Replay and Drift

We compare the performance of the current non-point based SOTA OOD detector in time-series data, i.e., Feng et al. (2021)'s detector with CODiT on vision temporal OODs. We use the same model architecture, $G_T$, and TTPE-NCM on replay and drift datasets as we use on CARLA dataset from Section 5.1.2.

**Replay Dataset.** Replay dataset is generated from the CARLA's 27 iD test traces on a clear day weather by randomly sampling a position in each trace. All images from the sampled position in the trace are replaced

---

[1]Automold is a software used for augmenting road images to have various weather and road conditions.

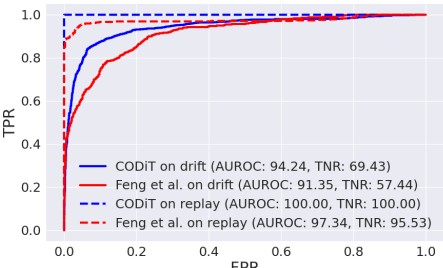 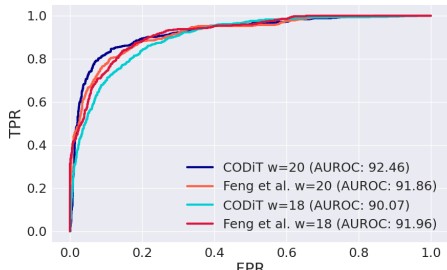

Figure 7: CODiT outperforms SOTA detector Feng et al. (2021) on temporal OODs in vision with the window length $w = 16$ (left). CODiT performs consistently well with different window lengths of $w = 18$, and 20 (right).

with the image at the sampled position in the original trace. Again, results are reported for $n = 20$, and on the sliding windows of replay OODs.

**Drift dataset.** We split 72 iD traces of cars driving straight without any drift from the drift dataset (Noor et al., 2020) into 24 for $X_{tr}$, 14 for $X_{cal}$, and 34 for test iD traces. Windows from $X_{tr}$ are sampled for training the VAE. Windows from $X_{cal}$ are sampled $n = 20$ times (with one window from one trace at a time to make each window in the set independent) for calculating the 20 sets of calibration non-conformity scores. We report results on the sliding windows of 34 iD test and 100 OOD drift traces.

**Results.** Fig: 7 (left) compares the ROC, AUROC and TNR (@95% TPR) results of CODiT with Feng et al. (2021)'s detector on the replay and drift datasets. We achieve SOTA results on both datasets.

**Ablations on drift.** We perform the following ablation studies on the drift dataset. All VAE models used in these studies are trained on the proper training set of the drift dataset with the same model architecture, $G_T$, and TTPE-NCM from Section 5.1.2.

(1) **Performance of CODiT with Different Window Lengths** $w$: We train two VAE models with $w = 18$, 20 and compare the performance of CODiT (with $n = 20$) with Feng et al. (2021)'s detector on these window lengths. Fig 7 (right) shows that with both $w = 18$ and 20, CODiT performs consistently well.

(2) **Performance of CODiT with Different $|\mathbf{G_T}|$:** We compare the performance of CODiT with different sizes of the transformation set. Table 3 shows that the performance of CODiT ($n = 5$) increases with $|G_T|$.

(3) **Using Deviation from iD $G_T$-equivariance as NCM:** With $G_T = \{2x\ speed, shuffle, reverse, periodic, identity\}$, and for all ground truth temporal transformations in $G_T$, Fig. 5 shows that the non-conformity score from TTPE-NCM is much higher for OOD windows than the test iD windows. This justifies our hypothesis that $G_T$-equivariance learned by a model on windows drawn from training distribution is not likely to generalize on windows drawn from OOD.

(4) **CODiT's Performance Increases with $n$:** We train three VAE models with $G_T$ equal to $\{2x\ speed, reverse, identity\}$, $\{2x\ speed, shuffle, periodic, identity\}$, and $\{2x\ speed, reverse, shuffle, periodic, identity\}$. Results on AUROC and TNR in Fig. 6 shows that the performance of CODiT improves with the number $n$ of $p$-values used for detection in all the three cases ($|G_T| = 3$, 4, and 5). This justifies our hypothesis that under one transformation an OOD window might behave as a transformed iD window but the likelihood of that decreases with the number of transformations.

(5) **Bounded False Detection Rate (FDR):** For the VAE model with $|G_T| = 5$ described above, we perform FDR experiments with a larger calibration dataset of approximately 862 calibration datapoints. This set is obtained by including all sliding windows on all calibration traces in the set. 34 calibration traces are randomly selected from the set of 48 in-distribution traces and the rest 14 are used as test traces. This is repeated 5 times and the generated box-plot of CODiT ($n = 5$) is shown in Fig. 6 (right). For all the values of $\epsilon = 0.05.k, k = 1, \ldots, 10$ in the plot, the average FDR is aligned with $\epsilon$.

### 5.2.2 Non-vision: GAIT

**Dataset.** We use the physiological sensory GAIT dataset (Hausdorff et al., 2000) for our case study on

Table 3: Performance of CODiT increases with the size of the transformation set $G_T$.

| $|G_T|$ | Transformations | AUROC |
|---|---|---|
| | Speed, Identity, Shuffle | 84.78 |
| 3 | Reverse, Shuffle, Identity | 85.47 |
| | Speed, Reverse, Identity | 87.67 |
| | Speed, Shuffle, Periodic, Identity | 88.08 |
| 4 | Speed, Identity, Shuffle, Reverse | 88.76 |
| | Speed, Reverse, Periodic, Identity | 89.56 |
| 5 | Speed, Shuffle, Reverse, Periodic, Identity | 90.78 |

Table 4: AUROC of baseline/CODiT for OOD detection on GAIT dataset with different window lengths $w$.

| OOD | $w = 16$ | | $w = 18$ | | $w = 20$ | |
|---|---|---|---|---|---|---|
| | Baseline | Ours | Baseline | Ours | Baseline | Ours |
| ALS | **78.25** | 68.62 | 77.73 | **79.61** | 77.99 | **80.69** |
| PD | 74.11 | **85.38** | 74.18 | **84.25** | 74.52 | **84.40** |
| HD | 76.97 | **94.17** | 76.64 | **95.42** | 76.68 | **93.74** |
| ALL | 76.23 | **83.85** | 75.99 | **86.66** | 76.21 | **86.68** |

non-vision temporal OODs. This dataset consists of records on 16 healthy control subjects. We split these into 6 for $X_{tr}$, 5 for $X_{cal}$, and 5 for test iD records. We use all 27 records from the severe patient group with neurodegenerative diseases as OOD records. These 27 records contain 9 records from each of the three diseases, namely Amyotrophic Lateral Sclerosis (ALS), Parkinson's (PD), and Huntington's disease (HD).

**Training Details, $G_T$, and TTPE-NCM.** We use the 1D derived time-series features from the dataset (the `.ts` files) to train a VAE model with the Lenet5 architecture (LeCun et al., 1998) on windows sampled from $X_{tr}$. Lenet5 uses 2D CNNs that can be used on the time-series data of 1D feature space. We use $G_T$ = {high-pass filter, high-low filter, low-high filter, identity}. By high-low (or low-high) filter, we mean that we apply high (or low)-pass filter to the first half features and low (or high)-pass filter to the last half features of the dataset. Again, we use the cross entropy-loss between the true and predicted transformation as the TTPE-NCM.

**Baseline and Results.** We use a one-class SVM trained on the auto-correlated features in the time dimension of all the sliding windows in $X_{tr}$ as a baseline. We report results on the sliding windows of the test iD and OODs records. Table 4 compares the AUROC performance of CODiT ($n = 100$) with baseline on individual and all (ALS, PD, and HD) OODs with different sliding window lengths $w$ (16, 18, and 20). As can be seen, CODiT outperforms the baseline except for one case.

# 6 Related Work

OOD detection in non time-series datasets such as CIFAR-10 (Krizhevsky et al., 2009) has been extensively studied and detectors with OOD scores based on the difference in statistical, geometrical or topological properties of the individual iD and OOD datapoints have been proposed. These detectors can be classified into supervised (Kaur et al., 2021), self-supervised (Tack et al., 2020), and unsupervised (Lee et al., 2017) categories. Unsupervised approaches do not require access to any OOD data for training the detector, while supervised approaches do. Self-supervised approaches are the current SOTA for OOD detection in non time-series data which require a self-labeled dataset for training the detector. This dataset is created by applying transformations to the training data and labeling the transformed data with the applied transformation. In this paper, we consider the problem of OOD detection in time-series data and the proposed approach, CODiT, is a self-supervised OOD detection approach, where the self-labeled dataset is created by applying temporal transformations on the windows drawn from the training distribution.

Recently, there has been interest in leveraging ICAD for OOD detection with guarantees on false detection rate (Cai & Koutsoukos, 2020; Ramakrishna et al., 2021; Kaur et al., 2022; Haroush et al., 2021). While iDECODe (Kaur et al., 2022), and Haroush et al.'s (2021) are OOD detectors for individual datapoints, Cai et al.'s (2020), and Ramakrishna et al.'s (2021) are detectors for time-series data. iDECODe (2022) uses error in the equivariance learned by a model with respect to a set of transformations on individual datapoints as the non-conformity measure in ICAD for detection in non time-series data. Haroush et al. 2021 propose using a combined $p$-value from different channels and layers of convolutional neural networks (CNN) for detection. It is not clear how to directly apply individual point detectors to time-series data with the ICAD guarantees due to the following two reasons. First, even if we apply these detectors to individual datapoints in the time-series window independently, we do not know how to combine detection verdicts on these datapoints for detection on the window. Second, for detection guarantees by ICAD, it is required that all non-conformity scores for $p$-value computation to be IID (Laxhammar & Falkman, 2015). Since these detectors are not solving OOD detection problem in time-series data it is not clear how to apply them to time-series while

preserving the IID assumption on the time-series data. Also, iDECODe uses a single $p$-value for detection and we propose using multiple ($n > 1$) independent $p$-values to be combined by the Fisher's method for preserving the detection guarantees. In contrast to (Haroush et al., 2021), our approach is not limited to CNN classifiers and can in principle be used for any type of other predictive models as well.

Cai et al. (2020) propose using reconstruction error by VAE on an input image as the non-conformity measure in the ICAD framework. Martingale formula (Vovk et al., 2003) is used to combine multiple $p$-values computed on multiple samples of the input in the latent space of the VAE. The detection score on a window is then computed by applying cumulative sum procedure (Basseville et al., 1993) on the martingale values of all the images in the window. Ramakrishna et al. (2021) propose using KL-divergence between the disentangled feature space of $\beta$-VAE on an input image and the normal distribution, as the non-conformity measure in the ICAD framework. They also use the martingale formula to combine $p$-values of all the images in the window for detection. Both these detectors are point-based, and as shown by the experiments on replay OODs, these might perform poorly in the detection of temporal OODs. CODiT computes the $p$-value of the window (and not individual datapoints in the window) in the ICAD framework. It is, therefore, a non-point based approach and as shown by experiments on temporal OODs in Section 5, it can be used to detect these OODs. To our knowledge, the current SOTA approach for OOD detection in time-series data is by Feng et al. (2021). They propose extracting optical flow information from a window of the time-series data and training a VAE on this information. KL-divergence between the trained VAE and a specified prior is used as the OOD score. This detector uses optical flow to extract time-dependency in the frames of a window and thus can be used to detect temporal OODs. However, this approach does not provide any guarantees on detection and will not work on non-vision datasets as it relies on optical flows. As shown in the experiments on the GAIT sensory dataset, CODiT can be used for OOD detection in non-vision datasets.

Anomaly detection in time-series data is also a closely related and an active research area (Ishimtsev et al., 2017; Guo & Bardera, 2018; Gao et al., 2020). In this paper, we consider the detection of a special class of anomalous data, the OOD data (data lying outside the training distribution). For instance, let us consider the case where most of the training data is clean and the rest is adversarially perturbed. In this case, the rare adversarial inputs are anomalous with respect to the training data, where most of the training data was drawn from the training distribution of clean data. However, adversarially perturbed inputs are not OOD as some of the training data was sampled from the training distribution of these adversarially perturbed data.

# 7 Conclusion and Discussion

We propose to use time-dependency between the datapoints in a time-series window for OOD detection on the window. Specifically, we propose using deviation from the temporal equivariance learned by a model on windows drawn from training distribution as an NCM in the conformal prediction framework for OOD detection in time-series data. Computing independent predictions from multiple conformal detectors based on the proposed measure and combining these predictions by Fisher's method leads to the proposed detector CODiT with guarantees on false OOD detection in time-series data. We illustrate the efficacy of CODiT by achieving SOTA results on computer vision datasets in autonomous driving, and the GAIT sensory data.

The time complexity of CODiT is as follows. At inference time, ICAD computes the non-conformity score of an input and compares it with the scores of the pre-computed (in offline settings) calibration datapoints for anomaly detection. The time-complexity of ICAD is therefore $\mathcal{O}$(non-conformity score computation of the input+|calibration set|). Non-conformity score computation in our case is the output generation (i.e., prediction of the applied transformation) by the VAE model. We found it to be approximately 0.003 seconds in our experiments. The time-complexity of ICAD is for calculating one $p$-value of the input. CODiT uses multiple ($n$) $p$-values and combine them using Fisher's test for OOD detection. So, the time-complexity of CODiT $= n \times$ time-complexity of the ICAD framework, where $n$ is the number of $p$-values. Therefore the time-complexity of CODiT increases linearly with the number $n$ of $p$-values used for detection. As seen from Fig. 6, detection performance of CODiT improves with $n$. So, it is a trade-off between time-complexity and detection performance. We also observe that the performance of CODiT improves with the number of transformations in the set of temporal transformations. So, using all (instead of choosing a subset) of temporal transformations suitable for the application works better for CODiT.

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

# A Appendix

## A.1 False Detection Rate Guarantees

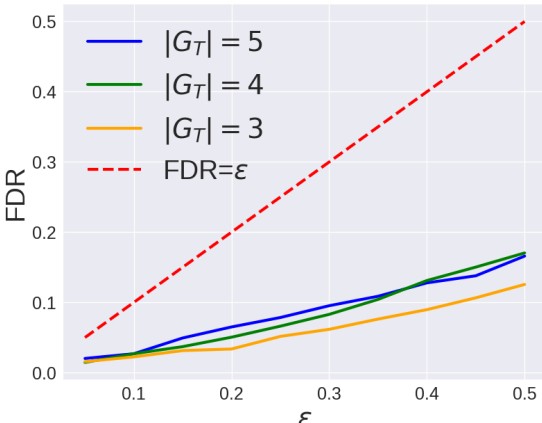

Figure 8: False Detection Rate (FDR) is much lower than $\epsilon$ for $|\text{calibration set}| = 14$ for all the three VAE models with $|G_T| = 3,\ 4,$ and 5.

With $\epsilon = 0.05 \cdot k, k = 1, \ldots, 10$, Fig. 8 shows that the false detection rate of CODiT ($n = 5$) is always less than $\epsilon$ for all the three VAE models ($|G_T| = 3,\ 4,$ and 5) described in the ablation studies on the drift dataset. The plots in Fig. 8 are reported with exactly one calibration window (sampled randomly) from each calibration trace in the drift dataset. This is because we want the calibration set to be IID for the FDR guarantees from the ICAD framework (Lemma 1) to hold. With 14 calibration traces in the drift dataset, we have exactly 14 calibration datapoints. Using a statistically insignificant number of 14 calibration datapoints gives us the false detection rates that are much lower than the detection threshold $\epsilon$.

The plots in Fig. 8 are reported with 14 calibration and 34 in-distribution test traces, totaling to 48 in-distribution traces. We increase the number of calibration datapoints to empirically check the FDR with respect to $\epsilon$. We increase the number of calibration traces from 14 to 34 and include all sliding windows on all the calibration traces in the calibration set. This gives us a calibration dataset with a larger number of approximately 862 calibration datapoints. 34 calibration traces are randomly selected from the set of 48 in-distribution traces and the rest 14 are used as test traces. This is repeated 5 times and the generated box-plot of CODiT is shown in Fig. 6 (right) of the paper. For all the values of $\epsilon = 0.05.k, k = 1, \ldots, 10$ in the plot, the average FDR is better aligned with $\epsilon$.

### A.1.1 Comparison with Baselines on False Detection Rate (FDR) with respect to the Detection Threshold $\epsilon$

**Comparison with the baselines**: Two of the existing baselines, i.e. VAE-based Cai et al.'s detector (2020), and $\beta$ VAE-based Ramakrishna et al.'s detector (2021) are also based on the ICAD framework. Since both of these approaches are point-based (i.e. treat each point in the window independently), we compare them with CODiT on CARLA's weather OODs. Fig. 9 shows the box plots for CODiT and Fig. 10 shows the box-plots for baselines on the CARLA dataset. Again, these box plots are reported on 5 trials with randomly sampled 27 calibration and 13 test traces from 40 in-distribution traces in each trial. Using all the sliding windows of the 27 calibration traces in this experiment gives us a total of approximately 2800 calibration datapoints.

**Observations**: The quartile range for Ramakrishna et al.'s method increases with $\epsilon$. For Cai et al.'s method, the average FDR is always much lower than $\epsilon$, i.e., average FDR is not calibrated with $\epsilon$. Average FDR for CODiT is better aligned with $\epsilon$ for all the values of $\epsilon$ in the plot and has a much lower quartile ranges than Ramakrishna's method.

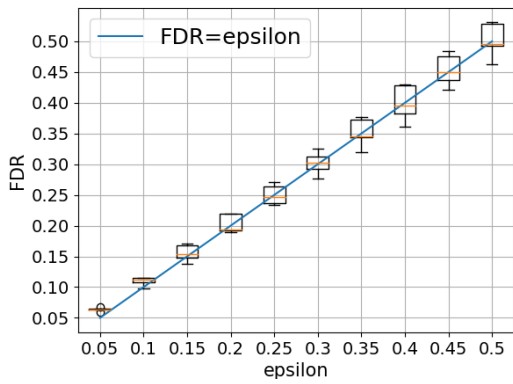

Figure 9: Box-plots on False Detection Rate (FDR) vs detection threshold $\epsilon$ for **CODiT** ($n = 5$) on the CARLA dataset.

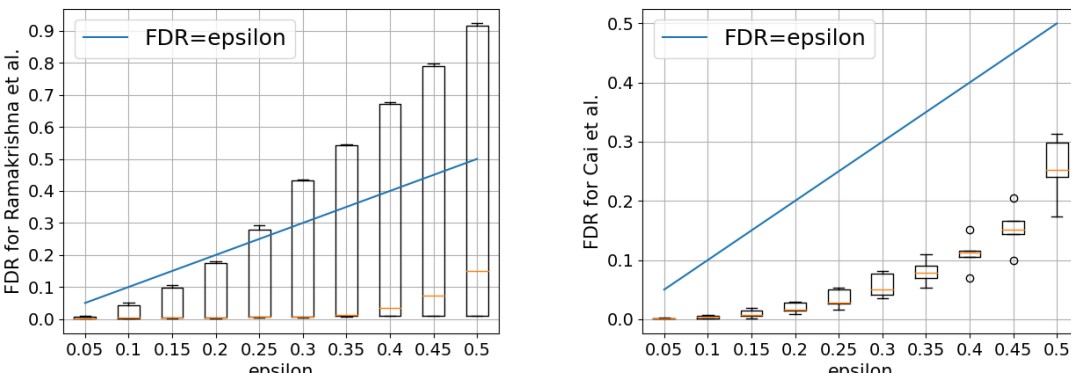

Figure 10: Box-plots on False Detection Rate (FDR) vs detection threshold $\epsilon$ for **baselines** on the CARLA dataset.

## A.2    Examples of iD and OOD windows

Here we show:

- A window from the iD trace of the drift dataset.

- A window from the iD trace of the CARLA dataset.

- Windows from the weather and night OOD traces from the CARLA dataset.

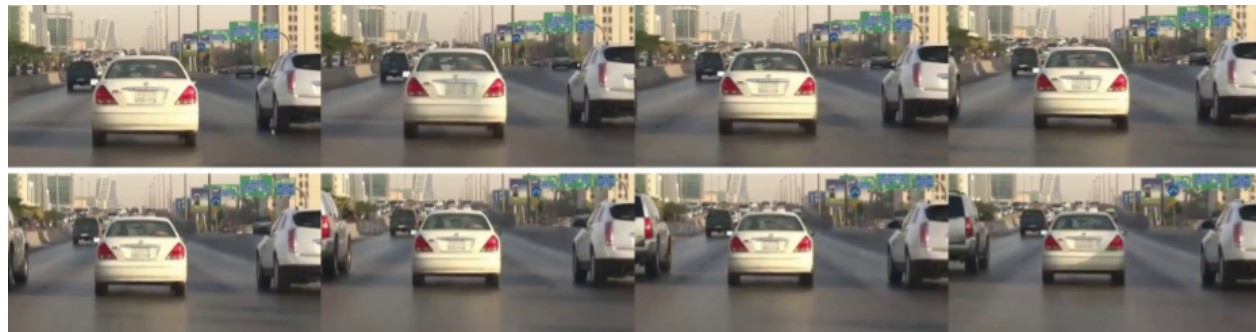

Figure 11: A window from an iD trace of the drift dataset: **car driving straight without any drift.**

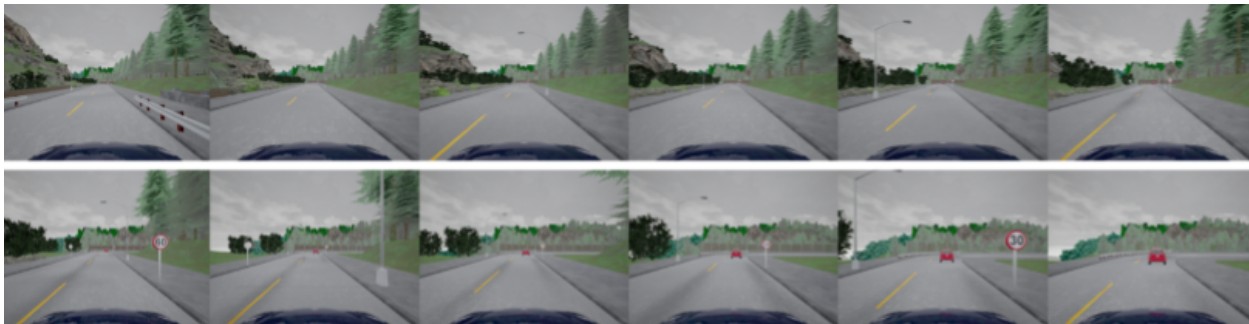

Figure 12: A window from an iD trace of the CARLA dataset: **driving in the clear daytime weather**.

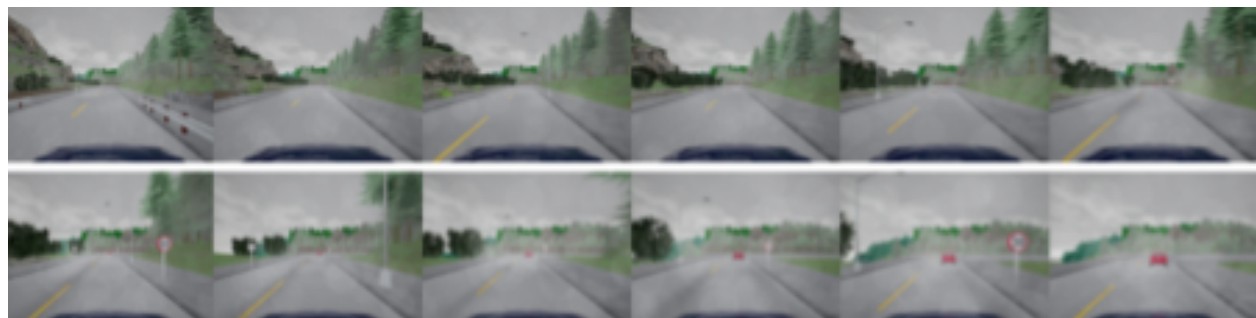

Figure 13: An OOD window from the **foggy** trace. The intensity of fog gradually increases in these OOD traces.

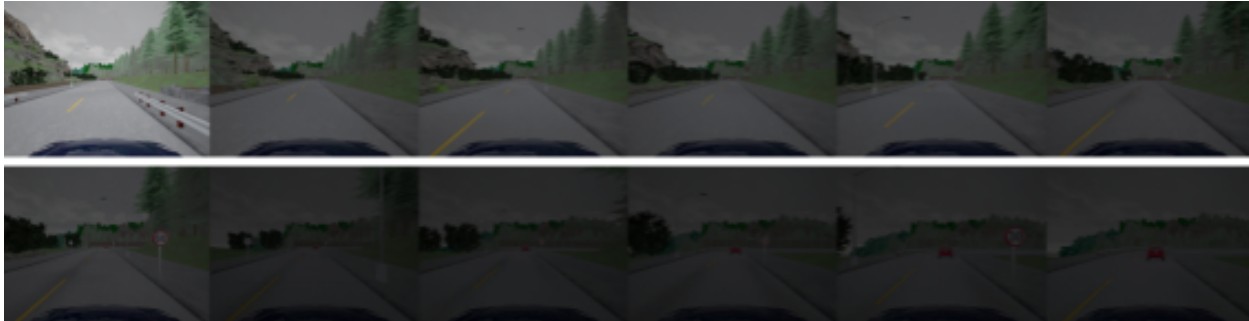

Figure 14: An OOD window from the **night** trace. The intensity of low brightness gradually increases in these OOD traces.

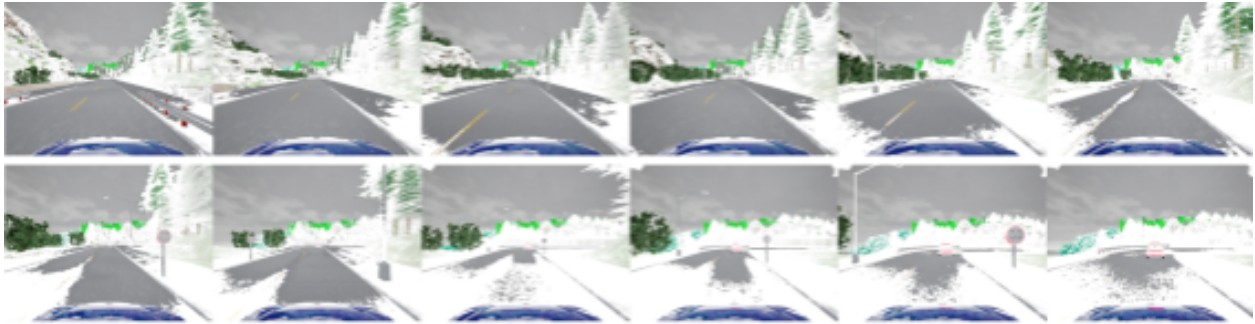

Figure 15: An OOD window from the **snowy** trace. The intensity of snow gradually increases in these OOD traces.

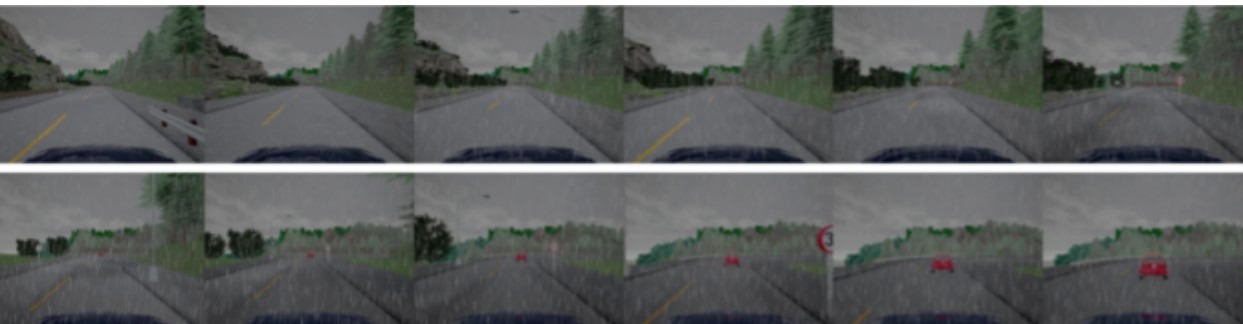

Figure 16: An OOD window from the **rainy** trace. The intensity of rain gradually increases in these OOD traces.

