# OpenReview forum: "CODiT: Conformal Out-of-Distribution Detection in Time-Series Data"
_TMLR — Rejected by TMLR_

### Review · Reviewer_CL2Q · 2022-06-13

**Summary Of Contributions:**

The authors propose an out-of-distribution detection method for time-series data using conformal prediction. They make two key contributions: 1. Training a VAE on windows from time series data (e.g., frames from a video) in an unsupervised fashion, taking into account temporal transformations such as reversing the time series, skipping frames, etc. 2. Using the reconstruction error of the VAE as a nonconformity score for inductive conformal anomaly detection (ICAD). The latter provides a guarantee on the false positive detection. Experimentally, they show that this approach is superior to related approaches, in particular when performing out-of-distribution detection on data where temporal information is crucial for the decision (in- or out-distribution).

**Broader Impact Concerns:**

There is no broader impact statement present, but I am also not concerned with ethical implications. This is mainly because I believe that making methods aware of in-/out-distribution is important for safety and any statistical guarantee on performance is very valuable.

**Requested Changes:**

Current conclusion:
I am not convinced that the paper is ready for being published at TMLR yet.

Main concerns and recommendations:
1. Address some of the comments on writing, especially regarding discussion of the guarantee, related conformal prediction approaches in the related work section, more meaningful conclusion/discussion and potentially better flow across sections 2 and 4.
2. Fix experiments by running trials with randomly sampled calibration sets in order to get appropriate estimates of false detection rates and OOD detection rates. I think this is critical as the guarantee holds marginally as far as I am aware.
3. Address why the actual false detection rate is much smaller than \epsilon – if this is still the case with random calibration sets.
4. Discuss why the temporal aspect is not a problem in obtaining the guarantee (as mentioned for other methods in related work). This seems to be critical and is not discussed at all.

**Strengths And Weaknesses:**

Strengths:
Form and writing:
- Clearly stated contributions, also putting the work in context of related methods.
- Examples in Figures 1 and 2 help motivate the approach.
- Elaborate background section providing details on equivariance, conformal prediction and Fisher’s method.
- Algorithm 1 summarizes the approach well.
- Thorough related work section.

Method:
- The approach tackles an important problem, how to perform out-of-distribution detection while taking into account temporal information in time series data.
- The guarantee of false positives is a nice application of conformal anomaly detection.

Experiments:
- Experiments on multiple datasets, with some settings specifically requiring temporal information for good decisions.
- Comparison to important baselines.

Weaknesses:
Form and writing:
- Section 2 feels misplaced. I would rather have it after related work as a motivation/introduction to the main sections.
- While related work is thorough (i.e., related work is discussed in detail), I feel that the authors should touch on conformal prediction more specifically. There are also some papers that I think are very relevant and should be discussed (even though they do not tackle time series data, the general idea of out-of-distribution detection with guarantees is similar):

[a] https://arxiv.org/abs/2102.12967
[b] https://openreview.net/pdf?id=Ro_zAjZppv

- The section on equivariance is a bit too verbose in my opinion. Space is better spent on a proper conclusion and discussion at the end.
- Section 4.2: The actual guarantee is only stated at the very end and not discussed at all. I would also find it important to discuss that this guarantee (i.e., the probability of a false positive) is marginal across calibration sets and not guaranteed conditional on the data.
- Section 4.3 is not really integrated in the main story (generally, section 4 could be written more compellingly): I think it should be emphasized why the reader needs to read this and how it will become important.
- Figure 4: this example/data is not well explained at all. What exactly am I looking at (esp. y-axis)? What does the data tell me?
- Conclusion is very short and in its current form not useful.
- Captions are generally very short and I would appreciate a bit more detail and maybe the take-away I am supposed to get from the figures.
- Table 2 could be reorganized to make it clearer which number corresponds to which methods – the separation via “/” does not work for me.

Method:
- Regarding the transformations: why should we use speed or shuffle? In Figure 5, the losses on OOD data seem actually lower than those without transformation (=identity). This seems weird to me. I assume that these losses are already on a model trained with these transformations?
- After theorem 1, there is discussion missing (e.g., that this is only marginally, unconditionally). Also, I think it should be discussed why the proof works: In the related work it is mentioned that it is unclear how to apply related work as p-values need to be IID. But it turns out that the application is actually straight-forward because we just assume IID? I think this is a critical point for time series data and it should be made clear why this works.
- The validity (i.e., false positive rate \epsilon) is empirically much smaller than what it was calibrated for. This suggests that theorem one is actually a loose upper bound. Usually, split conformal prediction is tight. I am wondering why this is? As I see it, theorem 1 is simply an application of Lemmata 1 and 2 (i.e., standard conformal prediction results), so it is unclear where this looseness comes from. Another option could be that the calibration set used is always very good (i.e., a lucky pick) – also see my comments on experiments.
- Comment on novelty and significance: Currently, I see not much novelty on the theoretical side, the obtained guarantee seems to follow very easily from existing work. Also, learning with temporal transformations as unsupervised task is not in itself novel. This means the key novelty of the work is the combination of both. I nevertheless appreciate the demonstration that this works well together, especially in cases where temporal information is needed.

Experiments:
- For the validity experiments, I would expect random trials across calibration/test splits. Especially for evaluating false detection rate I would expect to put calibration and test windows in one bucket and then draw multiple calibration and test sets randomly. Note that this also applies to the OOD performance. I think this is more or less standard in conformal prediction work (and should be lightweight as not re-training is involved).
- How does the false detection look for the baselines? Do some of the baselines also provide guarantees?
- In Figure 7, the blue problem seems very easy for the proposed method (compared to the baseline). Is there a way to consider a harder task to showcase some limitations or will the proposed method always work in this replay setting?

---

> ### Author Response · Authors · 2022-06-23
> **Addressing the concerns of the reviewer CL2Q**
>
> We appreciate and thank the reviewer for the thorough review. Here are our answers for addressing the concerns of the reviewer:
>
> Q- Regarding the transformations: why should we use speed or shuffle? In Figure 5, the losses on OOD data seem actually lower than those without transformation (=identity). This seems weird to me. I assume that these losses are already on a model trained with these transformations?
>
> Ans - Yes, these losses are from a model trained to predict an applied transformation from the set of temporal transformations = \{speed, shuffle, reverse, periodic, identity\}. In Figure 5, each plot is on randomly sampled 150 windows from the OOD traces and randomly sampled 150 windows from the test in-distribution traces. These windows are not the same across the plots.
>
> We do not know in advance which transformation will have higher loss on an OOD window and therefore propose to use multiple ($n$) transformations (sampled independently from the transformation set) on the input for detection. The intuition being that an OOD window might behave as an in-distribution window under one transformation (by having lower loss) but the likelihood of that decreases with the number of transformations.
>
> Q - After theorem 1, there is discussion missing (e.g., that this is only marginally, unconditionally). Also, I think it should be discussed why the proof works: In the related work it is mentioned that it is unclear how to apply related work as p-values need to be IID. But it turns out that the application is actually straight-forward because we just assume IID? I think this is a critical point for time series data and it should be made clear why this works.
>
> Ans:
> $\textbf{Discussion}$ - With $D$ as the training distribution, the unconditional probability that an input $X_{t,w} \sim D$ is classified as an OOD by Algorithm 1  is bounded by $\epsilon$ under the assumption that the calibration datapoints and $X_{t,w}$ are IID. For this guarantee to hold for a sequence of inputs, we require an independent calibration set for every input in the sequence. This is computationally inefficient for real-time applications and therefore a calibration set is fixed offline in the ICAD algorithm [1]. The average false detection rate on the sequence of inputs drawn from the training distribution in this setting  is expected to be calibrated with $\epsilon$ (or empirically even higher than $\epsilon$).
>
> $\textbf{Why the proof works}$ - If an input $X_{t,w}$ is sampled from the training distribution, then the non-conformity score of $X_{t,w}$ and the non-conformity scores of the calibration datapoints are IID conditioned on the proper training set and the set of temporal transformations. With $n$ non-conformity scores computed from $n$ transformations sampled independently from a distribution over the set of temporal transformations for both the input and the calibration datapoints, and all the calibration datapoints also sampled independently for each $n$, the $n$ $p$-values of $X_{t,w}$ are IID. Here, we require the transformations for both the input and the calibration datapoints to be sampled independently and also the $n$ sets of the calibration datapoints to be sampled independently for the $n$ $p$-values to be IID.
>
> $\textbf{Using IID calibration datapoints for time-series data}$ -  With calibration traces of varying lengths (and greater than the window length) in the experimental datasets, we consider exactly one calibration window per calibration trace for creating an IID calibration set for a $p$-value computation. This window is sampled randomly from the trace $n$ times to make the $n$ sets of calibration datapoints independent.
>
> Q - In Figure 7, the blue problem seems very easy for the proposed method (compared to the baseline). Is there a way to consider a harder task to showcase some limitations or will the proposed method always work in this replay setting?
>
> Ans - The proposed method will always work in the replay settings as it is impossible to predict the applied transformation on the replay windows, where all frames in the window are the same. This is because the output of each temporal transformation (be it speed, shuffle, periodic, reverse or identity) will always be the same on the replay window. Therefore, the prediction losses on these OOD windows will be much higher than these losses on the in-distribution (or normal sequence of time-series) windows.
>
> Q - Suggestions and requested changes on writing and form of paper.
>
> Ans - As per the suggestions of the reviewer, we are working on revising the paper and will post the revision as soon as possible.
>
> For questions on FDR, we post it in the next comment due to lack of space in this one.
>
> [1] Laxhammar, Rikard, and Göran Falkman. "Inductive conformal anomaly detection for sequential detection of anomalous sub-trajectories." Annals of Mathematics and Artificial Intelligence 74.1 (2015): 67-94.

---

> > ### Author Response · Authors · 2022-06-23
> > **Addressing FDR concerns of the reviewer CL2Q**
> >
> > Concern - Address why the actual false detection rate is much smaller than \epsilon – if this is still the case with random calibration sets.
> >
> > Ans:
> > $\textbf{Existing experiments}$ - The False Detection Rate (FDR) plot in Figure 6 of the paper is reported with exactly one calibration window (sampled randomly) from a calibration trace in the drift dataset. This is because we want the calibration set in a $p$-value computation to be IID for the theoretical FDR guarantees of the ICAD framework to hold.  With 14 calibration traces in the drift dataset, we have exactly 14 calibration datapoints. Using a statistically insignificant number of 14 calibration datapoints gives us the plot reported in the paper.
> >
> > $\textbf{New experiments}$ - The current plot in the paper is reported with 14 calibration and 34 in-distribution test traces, totaling to 48 in-distribution traces. To increase the number of calibration datapoints and empirically check the FDR with respect to $\epsilon$, we perform new experiments. We increase the number of calibration traces from 14 to 34 and include all sliding windows on all the calibration traces in the calibration set. This gives us a calibration set with a greater number of 862 calibration datapoints. As per the reviewer's suggestion, 34 calibration traces are randomly selected from the set of 48 in-distribution traces and the rest 14 are used as test traces. This is repeated 5 times and the box-plots of the FDR vs $\epsilon$ results are in Fig. 1 of this document: https://docs.google.com/document/d/1ltvGFZNem5IUzxioufNkUusG8oXWR1IXk4rg7KzVXdo/edit?usp=sharing. With higher number of calibration datapoints, the average FDR is aligned with $\epsilon$ for different values of $\epsilon$ in this box plot.
> >
> > Q - How does the false detection look for the baselines? Do some of the baselines also provide guarantees?
> >
> > Ans - Two of the existing baselines, ie. Cai et al.’s VAE-based approach [2] and Ramakrishna et al.’s $\beta$-VAE approach [3] are also based on the ICAD framework. Since both of these approaches are point-based (i.e. treat each point in the window independently), we compare them with CODiT on CARLA’s weather (or non-temporal) OODs. We generate box plots with 5 trials on randomly sampled 27 calibration and 13 test traces from 40 in-distribution traces in each trial. Again, we use all the sliding windows of the 27 calibration traces as calibration datapoints, which gives us a total of 2800 calibration datapoints in this experiment. Fig. 2 of this document: https://docs.google.com/document/d/1ltvGFZNem5IUzxioufNkUusG8oXWR1IXk4rg7KzVXdo/edit?usp=sharing, shows the FDR vs $\epsilon$ plots for the baselines. We also generate these plots for CODiT on CARLA dataset and report it in Fig. 3 of the document.
> >
> > While both the baseline are based on ICAD, the quartile range for Ramankrishna’s method increases with $\epsilon$. For Cai et al.'s method, the average FDR is always much lower the $\epsilon$ or not calibrated with $\epsilon$. FDR for CODiT is aligned with $\epsilon$ for all the values of $\epsilon$ in the plot and has much lower quartile ranges than Ramankrishna’s method.
> >
> > [2] Cai, Feiyang, and Xenofon Koutsoukos. "Real-time out-of-distribution detection in learning-enabled cyber-physical systems." 2020 ACM/IEEE 11th International Conference on Cyber-Physical Systems (ICCPS). IEEE, 2020.
> >
> > [3] Ramakrishna, Shreyas, et al. "Efficient Out-of-Distribution Detection Using Latent Space of β-VAE for Cyber-Physical Systems." ACM Transactions on Cyber-Physical Systems (TCPS) 6.2 (2022): 1-34.

---

> > > ### Comment · Reviewer_CL2Q · 2022-07-04
> > > **On additional experiments**
> > >
> > > I appreciate the additional experiments provided by the authors. Two comments on these results:
> > >
> > > 1. I am not sure whether I understand why the two baselines underestimate FDR so significantly. Is that because they are point-based and thus need to operate on significantly lower FDR while giving the same guarantee of epsilon?
> > > 2. I strongly believe that the experiments with random calibration/test splits should be the main results in the main paper. Keeping e.g. Figure 6 (right) as is misrepresents the approach since it suggests that the FDR guarantee is not met but underestimated, which is actually not the case considering random trials.

---

> > > > ### Author Response · Authors · 2022-07-05
> > > > **Addressing comments on experiments by Reviewer CL2Q**
> > > >
> > > > 1. I am not sure whether I understand why the two baselines underestimate FDR so significantly.....guarantee of epsilon?
> > > >
> > > > Ans. We agree with the reasoning provided by reviewer on the baselines for underestimating FDR. These baselines are $\textbf{point-based}$ and thus operate on lower FDR while giving the same guarantee of epsilon.
> > > >
> > > > 2. I strongly believe that the experiments...Keeping e.g. Figure 6 (right) as is misrepresents the approach since it suggests that the FDR guarantee is not met but underestimated, which is actually not the case considering random trials.
> > > >
> > > > Ans. We appreciate and thank the reviewer for helping us on improving the FDR results in the paper. We replaced the Fig. 6 (with FDR much lower than $\epsilon$) in the main paper with the box-plot of CODiT on drift dataset. We moved the Fig. 6 (with FDR much lower than $\epsilon$), along with the reasoning on lower FDR to the appendix.

---

### Review · Reviewer_tQq6 · 2022-06-18

**Summary Of Contributions:**

*Problem*: This paper addresses the problem of out-of-distribution (OOD) detection, i.e., detecting windows of data within a time series that are not drawn from the training distribution. Specifically, this paper focuses on detecting *temporal OODs*, which cannot be identified as OOD by simply looking at a single datapoint in isolation but rather are only OOD when considered as a sequence. An example of a temporal OOD is a video that gets stuck and continuously replays a single frame. Existing OOD-detection methods are not designed to identify temporal OODs or only work on vision datasets (e.g., Feng et al. 2021)

*Solution*: This paper builds upon a previous conformal anomaly detection framework (ICAD). Specifically, the contributions the authors make are:
1. They propose a new non-conformity score that can be applied to time series windows, thus allowing the framework to be used to solve the problem of OOD detection in time series.
2. Rather than computing a single p-value, they compute multiple p-values and then combine them via Fisher's method. They show in their ablation studies (Fig. 6) that computing multiple p-values rather than just one improves performance.

The authors also provide a theoretical guarantee on the false detection rate and experiments that demonstrate that their method performs favorably compared to existing methods.

**Broader Impact Concerns:**

No concerns.

**Requested Changes:**

- It would be good to provide some more intuition/a less jargony description of the proposed NCM in the Introduction.

- The current problem statement (Sec 2.1) could be clearer. As written, it seem like the windows all come from a single time series, but this seems to not be the case for some datasets, such as GAIT

- Section 5.1 is somewhat unclear. I had to read it a few times to understand what non-temporal OODs meant. I would suggest replacing the last two sentences of the first paragraph with "In contrast to the temporal OODs, the non-temporal OODs can be detected by looking at individual images. Examples of non-temporal OODs include..."  In general, it is confusing that "weather OODs" and "non-temporal OODs" are used interchangeably.

- There should be a more explicit description of how the calibration set $\alpha$'s are generated. Based on the way that the "OOD Detection Score" section on p. 8 is currently written, it seems as though the $n$ sampled transformations are all applied to the same window. It was not until I read the experimental setup section that I realized that the windows are also sampled.

Minor typos:
- p. 1: "has therefore gained quite attention" - remove "quite" or change to "quiet" (?)
- p. 3: Figure 2 caption "and generates same image" -> "and generates the same image"
- p. 6: Beginning of Sec 4.3 - "Same hypothesis" -> "The same hypothesis"
- p. 9: Proof - "the indepdently drawn" -> "the independently drawn"
- p. 10: "on a clear day weather" -> "in clear daytime weather"
- p. 11: "(4) CoDiT's Performance Increases with n:" - n should be $n$


**Strengths And Weaknesses:**

**Strengths**

- Simple and intuitive method with theoretical guarantee.

- Good experiments and ablations.

**Weaknesses**

- I have some confusion about the proof of Theorem 1. Is it true that the p-values are actually independent? For example, there is some non-zero probability that for two different p-value computations, we sample the same transformation and windows. This would imply that the p-values must be identical and thus not independent. If this is the case, Lemma 2 does not apply.

- It would be useful to get a sense of the computational efficiency of the proposed method.

- There are some places where the writing is unclear. See Requested Changes for more details.

---

> ### Author Response · Authors · 2022-06-23
> **Addressing concerns of the reviewer tQq6**
>
> We thank the reviewer for reviews and while we are working on revising the paper for addressing the concerns on the writing, here are our answers for the other concerns:
>
> Q - I have some confusion about the proof of Theorem 1. Is it true that the p-values are actually independent? For example, there is some non-zero probability that for two different p-value computations, we sample the same transformation and windows. This would imply that the p-values must be identical and thus not independent. If this is the case, Lemma 2 does not apply.
>
> Ans - We are not sure if we understand the concern of the reviewer, but here is our clarification on the independent $p$-values in CODiT. Transformation on the input from the transformation set, calibration window from the calibration trace, and transformation on the calibration window from the transformation set are all sampled independently for each $p$-value computation. Even though there is a non-zero probability that two $p$-values computed this way are identical, they are independent. For example, let us consider the case of sampling two datapoints from a uniform distribution independently. There is a non-zero probability these datapoints have the same value but that does not make these datapoints dependent.
>
> Q - It would be useful to get a sense of the computational efficiency of the proposed method.
>
> Ans - CODiT is based on an efficient version of the Conformal Prediction (CP) framework - the Inductive Conformal Prediction (ICP) framework. In ICP, a calibration set (disjoint from the training set) is sampled offline from the training distribution.  The non-conformity scores of the calibration set are calculated and stored before running the anomaly detection algorithm (Inductive Conformal Anomaly Detection or ICAD) on an input. At inference time, the non-conformity score of the input is calculated and compared with the pre-computed scores of the calibration set for anomaly detection. So, the run-time complexity for anomaly detection in ICAD is O(non-conformity score computation of the input+|calibration set|). Non-conformity score computation in our case is the output generation (i.e. prediction of the applied transformation) by the VAE model. We found it to be ~0.003 seconds in our experiments.
>
> The time complexity of ICAD is for calculating 1 $p$-value of the input. We propose using multiple ($n$) $p$-values and combining them using Fisher’s test for OOD detection. So, the complexity of CODiT = $n \times$complexity of the ICAD framework, where $n$ is the number of $p$-values. Therefore, the time complexity increases with the number $n$ of $p$-values. As seen from Figure 6 in the paper, detection improves with $n$. So, it is a tradeoff between the time complexity and detection performance.

---

### Review · Reviewer_Zg3a · 2022-06-21

**Summary Of Contributions:**

This paper looks to study how to do out-of-distribution detection for temporal data. Specifically, the authors posit that defining a non-conformity score in terms of the in-distribution temporal equivariance (over multiple transformations to observed time series data) can help not only do out-of-distribution detection but also assist in bounding the false detection rate. The authors provide some theoretical justification for their design decisions and many empirical results.

**Broader Impact Concerns:**

No concerns.

**Requested Changes:**

- Can you comment more explicitly on the necessary transformations G? How many transformations are needed for our estimates to concentrate? How brittle are the OOD estimates if we use "bad" or "insufficient" transformations? I would like the paper to describe this more explicitly, as it is an important design decision that is crucial to the successful use of CODiT in practice. The
- Stylistically, Section 3 feels like it is in the wrong place. I'd prefer this related work, which is important but does not help improve my understanding of Section 4/5, appear before the conclusion.
- Can you please complexity analysis of your proposed method? Sorry if I missed this. You suggest running conformal multiple times, but in practice running exact full CP can be computationally infeasible and outright intractable for some model classes. I would have liked to see a discussion regarding this.

**Strengths And Weaknesses:**

Strengths
- Well written paper that lays out guardrails for the paper's contents/contributions
- I really like the way Section 4 and 5 segue from each other!
- The experimental results seem to validate the proposed method, and their theory was easy to follow. Clever and clear use of Fisher's method.

Weaknesses
- I don't leave the paper convinced that CODiT will be my go-to time-series OOD detector: this doesn't discount the validity of the work. I just don't have the urge to play with the code of this method. The authors could do a better job of empirically selling the upside of CODiT without relying on the nice FDR guarantee.
- Other than that, I really enjoyed this paper, which is sound and correct, and expect it will excite many in the CP and OOD communities.

---

> ### Author Response · Authors · 2022-06-23
> **Addressing concerns of the reviewer Zg3a**
>
> We greatly appreciate the positive reviews by the reviewer. While we are working on revising the paper based on the feedback for writing, here are our answers for addressing the other concerns of the reviewer:
>
> Q - Can you please complexity analysis of your proposed method? Sorry if I missed this. You suggest running conformal multiple times, but in practice running exact full CP can be computationally infeasible and outright intractable for some model classes. I would have liked to see a discussion regarding this.
>
> Ans - CODiT is based on an efficient version of the Conformal Prediction (CP) framework - the Inductive Conformal Prediction (ICP) framework. In ICP, a calibration set (disjoint from the training set) is sampled offline from the training distribution.  The non-conformity scores of the calibration set are calculated and stored before running the anomaly detection algorithm (Inductive Conformal Anomaly Detection or ICAD) on an input. At inference time, the non-conformity score of the input is calculated and compared with the pre-computed scores of the calibration set for anomaly detection. So, the run-time complexity for anomaly detection in ICAD is O(non-conformity score computation of the input+|calibration set|). Non-conformity score computation in our case is the output generation (i.e. prediction of the applied transformation) by the VAE model. We found it to be ~0.003 seconds in our experiments.
>
> The time complexity of ICAD is for calculating 1 $p$-value of the input. We propose using multiple ($n$) $p$-values and combining them using Fisher’s test for OOD detection. So, the complexity of CODiT = $n \times$complexity of the ICAD framework, where $n$ is the number of $p$-values. So the time complexity increases with the number $n$ of $p$-values. As seen from Figure 6 in the paper, detection improves with $n$. So, it is a tradeoff between the time complexity and detection performance.
>
> Q - Can you comment more explicitly on the necessary transformations G? How many transformations are needed for our estimates to concentrate? How brittle are the OOD estimates if we use "bad" or "insufficient" transformations? I would like the paper to describe this more explicitly, as it is an important design decision that is crucial to the successful use of CODiT in practice.
>
> Ans - Ablation study on the size of the transformation set ($|G_T|$) in Table 3 of the paper shows that the performance of CODiT increases as we increase the number of transformations in $G_T$ from 3 to 5. So, we used all the five temporal transformations (speed, shuffle, reverse, periodic, identity) in our experiments for comparison with baselines.
>
> Although, we observe variation in the performance with different transformations in the set for $|G_T| = 3$, and $4$, but having more transformations in $|G_T|$ gives us better results. So, using all (without choosing a subset) the temporal transformations suitable for the application dataset works for CODiT.

---

> > ### Comment · Reviewer_Zg3a · 2022-07-12
> > **Some additional comments**
> >
> > Thank you for your response.
> >
> > Please add this complexity analysis wherever is most appropriate in the text, as I'm sure readers will appreciate it.
> >
> > Regarding the transformations, I was not referring to the number of transformations but rather the type of transformation. For example, if your set of transformations includes a few random transformations (e.g., meaningless perturbations or inversion), how much worse is the quality of your OOD estimate? I'm trying to understand how much the choice of transformation matters.

---

> > > ### Author Response · Authors · 2022-07-15
> > > **Addressing additional comments**
> > >
> > > 1. Please add this complexity analysis wherever is most appropriate in the text, as I'm sure readers will appreciate it.
> > >
> > > Ans. We have added this discussion in the last section (conclusion and discussion) of the paper.
> > >
> > > 2. Regarding the transformations, I was not referring to the number of transformations but rather the type of transformation. For example, if your set of transformations includes a few random transformations (e.g., meaningless perturbations or inversion), how much worse is the quality of your OOD estimate? I'm trying to understand how much the choice of transformation matters.
> > >
> > > Ans. To answer this question, we are running new experiments with transformations such as Gaussian blur, inversion, rotation etc. We will get back to this comment once we have the results.

---

> > > > ### Author Response · Authors · 2022-07-16
> > > > **Experimental results on performance of CODiT with different transformation set**
> > > >
> > > > We performed additional experiments to address the reviewer's comment on performance of CODiT with respect to the type of transformations.
> > > >
> > > > We train two AVT models to learn equivariance with respect to two sets of transformations  (of size 3 and 4) on the training traces of the drift dataset. We use the same experimental settings on the dataset as reported in the paper. The first set = {Gaussian blur, rotation by $90^\circ$, horizontal flip} and the second set = {Gaussian blur, rotation by $90^\circ$, horizontal flip, inversion}. We apply the same (randomly sampled) transformation from the set on all images in the window and predict the applied transformation on the window. The AUROC results are as follows:
> > > >
> > > > ---------------------------------------------------------------------------------------------------------------------------------------------
> > > >  |$\ \ \ \ \ \ \ \ \ \ \ \ \ \ \ \ \ \ \ \ \  \  \ \ \  \ $ Transformation set $ \ \ \ \ \  \ \ \ \ \ \ \ \ \ \ \ \ \ \ \ \ \ \ \  \ \ \  \ \ \ $                                             | $\  $ $n$  = 1 $\  $|$\  $    $n$ = 2 $\  $ |$\  $ $n$ = 3 $\  $|$\  $ $n$ = 4 $\  $|$\  $ $n$ = 5 |
> > > >
> > > > ---------------------------------------------------------------------------------------------------------------------------------------------
> > > >
> > > > |{Gaussian blur, rotation by $90^\circ$, horizontal flip}$\  \ \ \   \ \  \ \ \  \   \ \  \ \ \   \  \$ | 76.12 $\  \ $|$\  $ 77.25 $\  $|$\  $ 78.02 $\  $| 78.30 $\  $|$\  $ \$\textcolor{blue}{78.53}$ |
> > > >
> > > > -----------------------------------------------------------------------------------------------------------------------------------------
> > > >
> > > > |{Gaussian blur, rotation by $90^\circ$, horizontal flip, inversion}$\ $| 82.13 $\  \ $|$\  $ 86.84$\  $|$\  $ 88.33 $\  $| 89.19 $\  $|$\  $ $\textcolor{red}{90.03}$ |
> > > >
> > > > --------------------------------------------------------------------------------------------------------------------------------
> > > > These results are consistent with the ablation studies in the paper on:
> > > > 1) Increase in detection performance of CODiT with $n$ (no. of $p$-values used to compute Fisher-value for detection).
> > > > 2) Increase in detection performance of CODiT with the number of transformations in the transformation set.
> > > >
> > > > The reported AUROC results in the paper for CODiT ($n=5$) with $G_T$={Speed, Reverse, Identity} is $\textcolor{blue}{87.67}$ and with $G_T$ ={Speed, Reverse, Identity, Periodic} is $\textcolor{red}{89.56}$.
> > > >
> > > > These additional experiments show that temporal transformations give good OOD detection but CODiT is robust to the choice of transformations.

---

> > > > > ### Comment · Reviewer_Zg3a · 2022-07-16
> > > > > **Good Start**
> > > > >
> > > > > This is a good start but does not help practitioners select which transformations to add to the set.
> > > > >
> > > > > Please could you try the following two sets:
> > > > > A)  {identity, Gaussian blur, rotation by 90, horizontal flip}
> > > > > B)  {identity, identity, Gaussian blur, rotation by 90, horizontal flip, inversion}
> > > > >
> > > > > This should help us see if quality or mere quantity of transformation matters

---

> > > > > > ### Author Response · Authors · 2022-07-18
> > > > > > **Additional experiments with identity transformation**
> > > > > >
> > > > > > As requested by the reviewer, we perform additional experiments with identity transformation and compare it with the ones reported in the previous response:
> > > > > >
> > > > > > ----------------------------------------------------------------------------------------------------------------------------------
> > > > > > |$\ \ \ \ \ \ \ \ \ \ \ \ \ \ \ \ \ \ \ \ \ \ \ \ \ \ \ \ \ \ \ \ $ Transformation set $ \ \ \ \ \ \ \ \ \ \ \ \ \ \ \ \ \ \ \ \ \ \ \ \ \ \ \ \ \ \  \ \ \ \ \ \ \ $ | $\ $ $n$ = 1 $\ $|$\ $ $n$ = 2 $\ $ |$\ $ $n$ = 3 $\ $|$\ $ $n$ = 4 $\ $|$\ $ $n$ = 5 |
> > > > > >
> > > > > > ----------------------------------------------------------------------------------------------------------------------------------
> > > > > >
> > > > > > |{Gaussian blur, rotation by $90^\circ$, horizontal flip}$\ \ \ \ \ \ \ \ \ \ \ \ \ \ \ \ \ \ \ \ \ \ \ \ \ \ \ \ \ $ | 76.12 $\ \ $|$\ $ 77.25 $\ $|$\ $ 78.02 $\ $| 78.30 $\ $|$\ $ 78.53 |
> > > > > >
> > > > > > ----------------------------------------------------------------------------------------------------------------------------------
> > > > > >
> > > > > > |{Gaussian blur, rotation by $90^\circ$, horizontal flip, identity}$\ \ \ \ \ \ \ \ \ \ \ \ \ \ \ \ $ | 81.12 $\ \ $|$\ $ 84.71  $\ $|$\ $ 85.63$\ $| 86.40 $\ $|$\ $ 87.09 |
> > > > > >
> > > > > > ----------------------------------------------------------------------------------------------------------------------------------
> > > > > >
> > > > > > |{Gaussian blur, rotation by $90^\circ$, horizontal flip, inversion}$\ \ \ \ \ \ \ \ \ \ \ \ \ \ $| 82.13 $\ \ $|$\ $ 86.84$\ $|$\ $ 88.33 $\ $| 89.19 $\ $|$\ $ 90.03 |
> > > > > >
> > > > > > ----------------------------------------------------------------------------------------------------------------------------------
> > > > > >
> > > > > > |{Gaussian blur, rotation by $90^\circ$, horizontal flip, inversion, identity}$\ $| 82.07  $\ \ $|$\ $ 86.64$\ $|$\ $ 88.90 $\ $| 90.29 $\ $|$\ $ 90.84 |
> > > > > >
> > > > > > ----------------------------------------------------------------------------------------------------------------------------------
> > > > > >
> > > > > > The result for the transformation set size equal to $5$ (the last result) is consistent with the one reported in paper (with $G_T$={Speed, Shuffle, Reverse, Periodic, Identity}) as 90.78.
> > > > > >
> > > > > > ---------------------------------------------------------------------------------------------------------------------------------
> > > > > >
> > > > > > These results show that:
> > > > > >
> > > > > > 1) CODiT is robust to the choice of transformations in the transformation set.
> > > > > > 2) While performance of CODiT increases with the number of transformations, it converges at the set size of five.

---

### Author Response · Authors · 2022-06-29
**Submitting revised version of the paper**

We appreciate and thank the reviewers for a thorough review on the writing of the paper. We submit a revised version of the paper with all the changes $\textbf{highlighted in blue color}$. The paper satisfies the page limit of 12 pages after removing the striked off content from it.

The changes are as follows:

(1) As requested by the Reviewer tQq6

a) We fixed all the minor typos.

b) Computational efficiency of CODiT has been discussed in the last section (Conclusion and Discussion).

c) Providing more intuition/a less jargony description of the proposed NCM in the Introduction - Addressed it in the first contribution (proposed NCM) of the introduction.

d) Section 5.1 is somewhat unclear... are used interchangeably - Did the re-writing as per the reviewer's suggestion.

e) There should be a more explicit description of how the calibration set 's are generated - Added this description as the first paragraph of section 4.2.2, just before the description of CODiT's algorithm.

f) The current problem statement (Sec 2.1) could be clearer. As written, it seem like the windows all come from a single time series, but this seems to not be the case for some datasets, such as GAIT - We are not sure if we understand this concern of the reviewer. An input window do come from a single time-series. Each datapoint ($x_t, x_{t+1},\ldots$) in the time-series window can be multidimensional. In case of a video, the dimension of a datapoint in the window is the size of the image. In case of the GAIT dataset, the dimension of a datapoint in the window is 12 (with 12 features in the .ts file). The figure 4 on GAIT analysis compares the value of one of these 12 features, i.e. stride time for 5 continuous minutes of a healthy control and patients with neurogenerative diseases.

(2) As requested by Reviewer Zg3a

a) Can you comment more explicitly on the necessary transformations G?... successful use of CODiT in practice - We added a discussion on it in the last section of "Conclusion and Discussion".

b) Stylistically, Section 3 feels like it is in the wrong place. I'd prefer this related work, which is important but does not help improve my understanding of Section 4/5, appear before the conclusion - Moved the related work section before conclusion.

c) Can you please complexity analysis of your proposed method? - We added a discussion on it in the last section of "Conclusion and Discussion".

(3) As requested by Reviewer CL2Q

a) Section 2 feels misplaced.  - Moved the related work section (section 2) before conclusion.

b) Section 4.2 (\textbf{now section 3.2}): The actual guarantee is only stated at the very end and not discussed at all - Added a discussion on the unconditional guarantee after the proof of Theorem 1 in the paper.

c) Generally, section 4 (\textbf{now section 3}) could be written more compellingly - Added text in the starting of this section (now Section 3) to integrate into the story for the main technical section (now Section 4) of the paper.

d) Figure 4: this example/data is not well explained at all - Added explanation of the figure 4 in section 4.1.

e) Conclusion is very short and in its current form not useful - Modified the conclusion and added discussion to it.

f) Captions are generally very short...take-away I am supposed to get from the figures - Improved the captions of the figures in the paper.

g) Table 2...the separation via “/” does not work for me - Reorganized tables 2 and 4 to make them clearer by indicating which number corresponds to which methods.

h) Comments on FDR - Added a new section A.1 in Appendix for addressing the comments on FDR. This section includes discussion on the FDR Fig. 6 of the paper, new experiments (as reqeusted by the reviewer) with box plots of FDR wrt $\epsilon$, comparison of FDR guarantees with the baselines.

i) Discuss why the temporal aspect is not a problem in obtaining the guarantee..This seems to be critical and is not discussed at all - The main problem in getting ICAD guarantees for time-series data lies in the fact the datapoints in the same time-series are not IID (or even exchangeable). So, how to create an IID calibration set for computing a $p$-value for an input with FDR guarantees from ICAD? We added a paragraph on how we generate an IID calibration set in the starting of the section 4.2.2.

j) Also, I think it should be discussed why the proof works: In the related work it is mentioned that it is unclear how to apply related work as p-values need to be IID. But it turns out that the application is actually straight-forward because we just assume IID? - We do not assume $p$-values to be IID and require the $n$ IID calibration sets sampled independently from the calibration traces, $n$ non-conformity scores computed from $n$ transformations sampled independently from $Q_{G_T} for both the input and the calibration datapoints. We added this clarification in the proof of the Theorem 1.

Due to lack of space, we continue in the next comment.

---

### Author Response · Authors · 2022-06-29
**Submitting revised version of the paper (Continued)**

Addressing comments by Reviewer CL2Q:

1) Adding [a] https://arxiv.org/abs/2102.12967 [b] https://openreview.net/pdf?id=Ro_zAjZppv to the related work section on ICAD - We added discussion on [a] "A statistical framework for efficient out of distribution detection in deep neural networks" by Haroush et al. in the related work.

[b] While "TRACKING THE RISK OF A DEPLOYED MODEL AND DETECTING HARMFUL DISTRIBUTION SHIFTS" tries to distinguish between benign and harmful distribution shifts, as per our understanding it is not based on the Conformal Prediction framework. They utilize sequential estimation to develop tests that provably control the false alarm rate.  They argue why Conformal Prediction does not perform the job of differentiating between benign and harmful shifts (Appendix A.2).

---

### Author Response · Authors · 2022-07-05
**Submitting revised version**

We submit a revised version of the paper after:

1) Removing the striked off content from the paper.
2) Changing the color of blue text to black.
3) Moving the Fig. 6 (with FDR lower than $\epsilon$) and its explanation to Appendix A.1.
4) Moving CODiT's box-plot of FDR with respect to $\epsilon$ on the drift dataset to Fig. 6 (right) of the main paper. We also modify the ablation section of the main paper on "(5) Bounded False Detection Rate (FDR):" to explain the box-plot.

---

### Decision · Action_Editors · 2022-08-07

**Recommendation:** Reject

**Comment:**

The authors develop a novel method for detected out of distribution sequences in temporal data, with theoretical guarantees that are based on the theory of conformal prediction. The reviewers agree that the method is sound both theoretically and in terms of the experimental methodology, and that the quality of writing is acceptable. However, reviewers had important concerns regarding how the claims made in the paper are substantiated by the results in the paper. In particular:

1. The theoretical guarantees require that multiple independent calibration sets are available, which is an unrealistic assumption in most practical scenarios. Further, the difficulty of establishing the theoretical guarantee is over-emphasized and is a straightforward application of the theory of conformal prediction to temporal data.

2. The method and its effectiveness relies on the user developing transformations to apply, which limits the utility of the approach as an off-the-shelf method unlike standard conformal prediction which can be applied to any black box predictor.

3. Evaluated purely empirically, it is unclear how the results improve upon prior work on OOD detection for temporal data.

I would encourage the authors to prepare a substantial revision adjusting the claims made. In particular, I would suggest that the author address:
1) The significance of the theoretical contributions, making precise the new contributions made versus what follows from standard results.
on conformal prediction.
2) How realistic the assumptions required for the theoretical results are (in particular having multiple iid calibration sets).
3) Explaining how a user of the approach should choose the transformations required.
4) Comparing against other OOD detection methods for temporal data including those that do not come with theoretical guarantees, or revising the language in the paper claiming "SOTA" empirical results.